# Managerial overconfidence in capital structure decisions and its link to aggregate demand: An agent-based model perspective

**Marcin Rzeszutek** [1⊕‡]*, **Antoine Godin**[2,3⊕‡], **Adam Szyszka**[4], **Stanislas Augier**[2]

**1** Faculty of Psychology, University of Warsaw, Warsaw, Poland, **2** Centre d'Economie de l'Université de Paris Nord (CEPN), Villetaneuse, France, **3** Institut für Makroökonomie und Konjunkturforschung (IMK) Hans-Böckler-Stiftung, Düsseldorf, Germany, **4** Department of Capital Markets, Warsaw School of Economics, Collegium of World Economy, Warsaw, Poland

⊕ These authors contributed equally to this work.
‡ These authors share first authorship on this work.
* marcin.rzeszutek@psych.uw.edu.pl

**Data Availability Statement:** All relevant data are within the paper and its Supporting Information files.

## Abstract

### Objective

This study aims to connect two strands of the psychology and economics literature, i.e., behavioural finance and agent-based macroeconomics, to assess the impact of managerial overconfidence at the micro and macro levels of the economy as a whole.

### Method

We build a macroeconomic stock-flow consistent agent-based model that is calibrated for the specific case of Poland to explore whether the overconfidence of top corporate managers in the context of their initial capital structure decisions is detrimental for the firms being managed in this way, the financial market dynamics, and the selected macroeconomic indicators. We model heterogeneous firms with different capital structure decision criteria depending on their degree of managerial overconfidence. Our model also includes a complete macroeconomic closure with aggregated households, capital producers, banking, and a public sector.

### Results

We find that firms with overconfident managers outperform in terms of investment and size but are also more fragile, thereby making them more likely to default. Finally, we run policy shocks and show that while investors' flight to liquidity creates financial turmoil and increases the probability of default.

### Conclusions

This paper contributes to the knowledge base by linking behavioural corporate finance and agent-based macroeconomics. In general, the excess overconfidence on the micro level, either an increase in the proportion of overconfident firms or a higher degree of

**Funding:** The study was financed from the funds of the National Science Centre in Poland under grant no. 2016/23/B/HS4/01283.

**Competing interests:** The authors have declared that no competing interests exist.

overconfidence among managers, has a strong destabilizing impact on the economy as a whole on the macro level.

## 1. Introduction

Since the beginning, behavioural finance proponents have been greatly inspired by psychology in creating a real picture of investors and of the stock market as a whole (e.g. [1–3]). A huge amount of research in cognitive psychology (see [4] showing that humans have limited cognitive abilities and that their decisions depend primarily on emotions has highlighted the need to modify several unrealistic assumptions in the area of neoclassical finance (see, e.g. [5, 6]), particularly regarding fully rational investors and efficient capital markets. Specifically, behavioural finance authors have described many behavioural biases among investors arising from cognitive errors and heuristics that result in stock market anomalies and asset mispricing [3, 7, 8], which has seriously challenged neoclassical finance. In the end, the behavioural school is often treated as a new paradigm in economics [3, 9]. However, there are still some research gaps in the behavioural finance literature that require further research. One of these relates to behavioural corporate finance, which deals with bounded rationality and its market consequences among corporate managers [10, 11].

Neoclassical corporate finance has made several assumptions to describe rational corporate managers who operate on efficient market and always seek to maximize shareholders' wealth (e.g. [12]). One of the most important issues in this regard is managerial decisions concerning the target capital structure of a company. Miller and Modigliani [13]'s theorem underlined that in the perfect capital market the value of a company is irrelevant to the way it is financed. In other words, the value of the firm stems simply from its revenue streams and the extent of related market risk and thus depends neither on decisions to raise equity via stock versus debt nor on the company's dividend policy. In addition, when the perfect capital market assumption is waived and corporate taxes are taken into account, the trade-off model assumes that managers always search for the rational mix of debt and equity that reflects optimal balance between the tax advantages of debt (tax shield) and costs related to insolvency [14–16]. According to the standard pecking order theory [17], managers always prioritize the sources of firm financing in the following order: 1) internal funds, 2) debt, and 3) new equity. This order is the result of the asymmetry of information between managers and shareholders. Managers are better informed about the firm's current and future financial situation than shareholders and always act in the firm's best interest, assumptions that have been undermined even by authors supporting the classical approach in the finance literature (e.g. [18, 19]).

The behavioural corporate finance literature provides much more empirical evidence against traditional theories on the target capital structure, relating it to one behavioural bias among corporate managers—overconfidence. In particular, overconfident managers overestimate the value of a company's investment possibilities and perceive the company stock to be undervalued. As a consequence, they are usually strongly against issuing new equity, subjectively assess debt financing as too costly, and therefore rely only on internal funds [20, 21]. In recent years, Malmendier, Tate, and Yan [22] found that overconfident managers overestimate future cash flows and thus consider external financing and equity in particular to be very costly. These authors additionally observed that managerial overconfidence was positively associated with debt conservatism and heightened leverage ratios. Therefore, it seems the standard pecking order theory reflects more managerial overconfidence than the above-mentioned asymmetry of information [23]. Ben-David, Graham, and Harvey [24] observed that

overconfident managers are less keen to pay dividends and more eager to repurchase shares. Finally, managerial overconfidence may be linked with a tendency to choose more short-term financing, in particular a preference for risky short-term debt over equity [25].

Apart from the aforementioned internal determinants of capital structure, there is considerable research on various external predictors of the target capital structure in the form of macroeconomic indicators [26–28]. Several studies have shown a significant association between corporate capital structure and the current gross domestic product (GDP) (e.g. [29, 30]), the inflation rate [31] the stock market and banking sector development [32, 33] and even the unemployment rate [34]. However, these studies concentrated only on the one-way direction of this relationship, that is, how these macroeconomic indicators affect managerial decisions regarding the capital structure. Much less examined is how the target capital structure can impact the macro level. Therefore, our main goal here is to examine the interconnection between managerial decisions regarding capital structure at the company level and macroeconomic aggregate demand using a macroeconomic agent-based modelling.

Agent-based modelling is a relatively new simulation method that links the micro and macro (or aggregate) parts of the analysis, a methodological approach that is lacking in behavioural finance and economics (see, e.g. [35–40]). In ABM, artificial financial markets are composed of heterogeneous, bounded rational agents with different expectations and behaviour and may represent a new theoretical paradigm in economic thought by providing a coherent and exhaustive representation of the inter-linkages between the financial and real sides of economic systems [41]. This simulation tool is especially helpful in the specific case of our study where we want to understand how managerial decisions will be affected and will affect managerial counterparts on the macro level (i.e. banks, investors, government). Specifically, corporate managers often have to face at least three contradictory motives in their decisions regarding firm financing [10, 11]. First, they need to fuel market share. To achieve this, they need to invest by accumulating physical capital, by buying out competitors, or by investing in innovation. Second, they are obliged to satisfy their shareholders; that is, they need to distribute dividends or buy back shares. Third, they have to satisfy bankers by offering satisfactory interest or making sure that leverage is at the appropriate level. If managers fail to meet one or more of these goals, the firm will be in trouble, which may also have consequences at the macro level.

There are two types of agent-based models (ABMs)—sectorial models that aim to analyse specific behaviours within a well-defined sector, such as a financial market (e.g. [38, 42–44]), and macroeconomic models [45–47] that focus on describing how different markets, each composed of agents, interact with each other. In our study, we aim to analyse macroeconomic effects and therefore concentrate on the second type of ABM. This category can be divided into partial and comprehensive models. Partial models typically concentrate on specific sectors and use macroeconomic dummies to represent neglected aspects. An interesting strand of the literature on banking/financial ABMs has emerged regarding the seminal model presented by Greenwald and Stiglitz [48]. This model originally highlighted the diffusion of idiosyncratic shocks within banking and commercial networks, leading to aggregate business cycles. It has recently been supplemented to include the concept of bankruptcy avalanches, business cycles [49], financial accelerator dynamics [50], long-lasting credit networks [51], and financial market accelerator dynamics [52].

Financial dynamics have been widely analysed using comprehensive macro agent-based models (see, e.g. [45, 47, 53, 54]). One main limitation of such studies is that they concentrate on the financial fragility emerging out of the banking sector via inequality [52, 55] monetary union imbalances [56, 57], banking regulations (or the lack thereof), and banking behaviours [45, 47, 54]. None of these contributions explicitly consider financial market trading equity.

Therefore, the current paper aims to develop a category of model between the partial and the comprehensive models. In line with some other studies (e.g. [58–62]), we developed a comprehensive model where some sectors remain aggregated while others are disaggregated. This avoids having to rely on macroeconomic dummies while keeping the complexity of the model reasonable.

## 1.1. Current study

This study aims to connect two strands of the psychology and economics literature, i.e., behavioural finance and agent-based macroeconomics, to assess the impact of managerial overconfidence at the micro and macro levels of the economy as a whole. Therefore, we created an ABM to investigate whether overconfidence in decisions regarding firm financing by chief executive officers (CEOs) and chief financial officers (CFOs) of companies listed on the Warsaw Stock Exchange (WSE) impacts the firms and whether it is associated with aggregate demand, that is, consumption, investment, and government expenditure. To the best of our knowledge, no study on this association exists in the literature on behavioural corporate finance or agent-based modelling. Additionally, our model is calibrated for the specific situation of Poland and uses the findings of the Rzeszutek and Szyszka [63] survey and selected Polish macroeconomic indicators. The remainder of the paper is structured as follows. Section 2 describes the structure of the model, Section 3 covers consumption firms, Section 4 covers the rest of the economy, Section 5 describes the baseline results, Section 6 discusses policy shocks, and Section 7 consists of the conclusion.

## 2. Structure of the model

Our aim is to understand the impact of managerial overconfidence regarding their individual future performances on micro and macro dynamics. Specifically, we assess how these corporate behavioural biases can affect firms' liability structure, investment decisions, and growth prospects. From a macroeconomic perspective, we are interested in their impact on financial stability, stock market volatility, and investment.

### 2.1. Accounting structure

To achieve our aim, we build an agent-based stock-flow consistent (AB-SFC) model the aggregate structure of which is outlined by the balance sheet (BS; Table 1) and the transaction flow matrix (TFM; Table 2). The BS allows one to see the financial structure of an economy where, except for capital goods and inventories, each row indicates the asset holder with a '+' and the liability emitter with a '-'. For example, one can see that loans are an asset for banks and a

**Table 1. Balance sheet.**

| | Households | C-Firms | K-Firms | Banks | Government | Total |
|---|---|---|---|---|---|---|
| [*Capital*] | | $+p \cdot K$ | | | | $+p \cdot K$ |
| [*Inventories*] | | $+p \cdot Inv$ | | | | $+p \cdot Inv$ |
| Deposits | $+D_h$ | $+D_f$ | | $-D$ | | 0 |
| Reserves | | | | $+R_b$ | $-R_b$ | 0 |
| Loans | | $-L$ | | $+L$ | | 0 |
| Bonds | $+B_h$ | | | $+B_b$ | $-B$ | 0 |
| Traded Equity | $+EQ_c$ | $-EQ_c$ | | | | 0 |
| Private Equity | $+EQ_b$ | | | $-EQ_b$ | | 0 |
| Total | $NW_h$ | $NW_f$ | 0 | $NW_b$ | $NW_g$ | $NW_{tot}$ |

**Table 2. Transaction-flow matrix.**

| | Households | C-Firms | | I-Firms | Banks | Government | Total |
|---|---|---|---|---|---|---|---|
| | | Current Ac. | Capital Account | | | | |
| Consumption | $-C_h$ | $+C_h$ | | | | | 0 |
| Government Expenditure | | $+C_g$ | | | | $-C_g$ | 0 |
| Inventories | | $+p \cdot \Delta Inv$ | $-p \cdot \Delta Inv$ | | | | 0 |
| Investment | | | $-p \cdot I_k$ | $+p \cdot I_k$ | | | 0 |
| Wages | $+W \cdot (E_c+E_k+E_g)$ | $-W \cdot E_c$ | | $-W \cdot E_k$ | | $-W \cdot E_g$ | 0 |
| Taxes | $-(T_w+T_f)$ | $-T_c$ | | $-T_k$ | | $+T_{tot}$ | 0 |
| Interest on Loans | | $-i_L \cdot L$ | | | $+i_L \cdot L$ | | 0 |
| Interest on Deposits | $-i_d \cdot D_h$ | $+i_d \cdot D_f$ | | | $-i_d \cdot (D-OF_b)$ | | 0 |
| Interest on Bonds | $+i_b \cdot B_h$ | | | | $+i_b \cdot B_b$ | $-i_b \cdot B$ | 0 |
| Dividends | $+Div$ | $-Div_c$ | | $-Div_k$ | $-Div_b$ | | 0 |
| Retained Earnings | | $-\Pi_u$ | $+\Pi_u$ | | | | 0 |
| | $[S_h]$ | | $[S_c]$ | 0 | $[S_b]$ | $[S_g]$ | 0 |
| Deposits | $-\Delta D_h$ | | $-\Delta D_f$ | | $+\Delta D - \Delta OF_b$ | | 0 |
| Loans | | | $+\Delta L$ | | $-\Delta L$ | | 0 |
| Equity | $-P_{eq,-1} \cdot \Delta Eq_c$ | | $+P_{eq,-1} \cdot \Delta Eq_c$ | | | | 0 |
| Bonds | $-\Delta B$ | | | | | $+\Delta B$ | 0 |
| Total | 0 | 0 | 0 | 0 | 0 | 0 | 0 |

liability for consumption firms (C-firms). The TFM allows one to see all the transactions taking place in the economy, be they non-financial (higher part of the TFM) or financial (lower part of the TFM). In terms of the BS, each row of the TFM has an outflow sector indicated with a '-' and an inflow sector indicated with a '+'. The consumption row thus can be understood as households buying goods from C-firms. The BS and the TFM highlight the strict accounting framework used in this paper by noting that each row of the matrices sum to zero. This accounting framework is at the heart of the SFC literature; see Godley and Lavoie [64] for a detailed description of the approach and Caverzasi and Godin [41] and Nikiforos and Zezza [65] for literature surveys.

The model consists of a closed economy with five sectors—households, consumption goods producers, capital goods producers (K-firms), private banks and a government. C-firms produce consumption goods sold to households and the government, K-firms provide investment goods to C-firms, banks grant loans to C-firms and hold government bonds and the government raises taxes and issues bonds to finance its expenditures. Finally, households consume and allocate their savings between deposits, government bonds and firms' shares. We explicitly model a stock market where firms can issue equity to households and buy it back. However, for the sake of simplicity we assume that K-firms and banks are privately owned. We microfound the C-firms sector, while the rest of the economy is modelled at the aggregate level.

### 2.2. Period structure

Each simulation period represents one quarter. Over the course of each period, the following sequence of events takes place:

1. K-firms set the price of investment goods.

2. C-firms compute demand and profit expectations and plan production and investment accordingly. C-firms also set their prices and make portfolio adjustments, issuing or buying back debt and/or equity.

3. K-firms produce investment goods on demand and sell them to C-firms.

4. Households determine their demand for consumption goods and allocate their savings. The government determines its expenditures budget. Firms' equity price is determined by households' portfolio decisions. Nominal wage for the next period is determined. Banks adjust their interest rate and their bonds holding.

5. Aggregate demand for consumption goods is allocated between C-firms. C-firms then compute their realized profits and pay dividends and interest. The government collects taxes and issues bonds that are bought by households.

6. C-firms' bankruptcy conditions are checked, and defaulted firms are replaced by new ones following the process described below.

## 3. Consumption firms

### 3.1. Production and price

C-Firms' expected demand ($yd_i^e$, 1) follows a double adjustment process; it combines a linear correction, using a constant parameter $\lambda_{yd_i^e}$, to past discrepancies between realized ($yd_i$) and expected demand with a trend based on firms' expected growth of demand. This growth anticipation is a weighted sum, with constant parameter $\lambda_g$, of observed average growth for the last four periods (corresponding to a year) at the firm ($\overline{g}_i$) and the macroeconomic levels ($\overline{g}$). C-firms have a target inventories to expected demand ratio ($\theta_{inv}$, 2), following [66, 67]. C-firms production ($y_i^p$, 3) is defined by expected demand plus supplementary production aimed at adjusting inventories ($v_i$) towards the desired level ($v_i^T$), with adjustment speed $\lambda_{inv}$. C-firms expected sales are bounded by their production decision plus available inventories ($s_i^e$, 4):

$$yd_i^e = [\lambda_{yd_i^e} \cdot yd_{i,-1} + (1 - \lambda_{yd_i^e}) \cdot yd_{i,-1}^e] \cdot [1 + \lambda_g \cdot \overline{g}_i + (1 - \lambda_g) \cdot \overline{g}] \tag{1}$$

$$v_i^T = \theta_{inv} \cdot yd_i^e \tag{2}$$

$$y_i^p = \max\left\{\frac{k_i}{v}, yd_i^e + \lambda_{inv} \cdot (v_i^T - v_{i,-1})\right\} \tag{3}$$

$$s_i^e = \min(yd_i^e, y_i^p + v_{i,-1}) \tag{4}$$

C-firms produce using a Leontief production function where labour is the abundant factor. Consequently, they cannot produce more than their full capacity output. The number of workers ($e_i$, 5), the rate of capacity utilization ($u_i$, 6) and the unit cost of production ($uc$, 7) are defined following the usual equations. For the sake of simplicity, we ignore innovations dynamics so that all C-firms use the same production function, with labour productivity ($A$) that grows at an exogenously given rate and capital-to-output ratio ($v$). The nominal hourly wage ($w$) is the same across all C-firms so that unit cost ($uc$) is also uniform:

$$e_i = \frac{y_i^p}{A} \tag{5}$$

$$u_i = \frac{y_i^p \cdot v}{k_i} \tag{6}$$

$$uc = \frac{w}{A} \tag{7}$$

C-firms have a desired price level ($p_i^T$, 8) defined by a mark-up ($\mu_i$, 9) over unit costs, see Lee [68] for more details on various pricing theories. The mark-up is an endogenous function of utilization and firms' share of demand ($f_i$). Due to frictions, and similar to Calvo [69], firms have a probability ($prob_i^p$, 10) of being able to adjust their price ($p_i$, 11) at each period that depends on the discrepancy between their current and target price levels:

$$p_i^T = (1 + \mu_i) \cdot (1 + \theta_y) \cdot uc \tag{8}$$

$$\mu_i = \mu_0 + \mu_1 \cdot u_i + \mu_2 \cdot f_i \tag{9}$$

$$prob_i^p = min\left(\lambda_p^0 + \lambda_p^1 \cdot |\frac{p_i - p_i^T}{p_i^T}|, 1\right) \tag{10}$$

$$p_i = \begin{cases} \lambda_p \cdot p_i^T + (1 - \lambda_p) \cdot p_i & \text{if } \epsilon_{i,p} < prob_i^p \\ p_{i,-1} & \text{else} \end{cases}, \tag{11}$$

where $\epsilon_{i,p}$ is the realization of a uniform distribution on the [0; 1] interval.

## 3.2. Expectations and investment

A specificity of this paper is that we introduce two types of C-firms. Firms belonging to the first group, classified as neutral firms, compute their expected profits ($\pi_i^e$, 12) based on their expenditures (composed of wages, interests payments on debt -$i_{l,i} \cdot l_{i,-1}$ – and interests receipts on deposits -, $i_d \cdot d_{i,-1}$-) and expected demand. The other group of firms, called overconfident firms, tend to overestimate their future profits. To represent this, we multiply their profit expectations by a random term ($\epsilon_{i,\pi}$, 13). C-firms then compute how much dividends they expect to distribute ($div_i^e$, 14) based on their expected profits and a constant saving rate ($s_f$); overconfident firms thus transmit the bias in their profit expectations to their dividend expectations, leading to an overestimation of the cost of equity, as we see below:

$$\pi_i^e = \begin{cases} \frac{p_i}{1 + \theta_y} \cdot s_i^e - w \cdot e_i - i_{l,i} \cdot l_{i,-1} + i_d \cdot d_{i,-1} & \text{if } i \in \{\text{Neutral}\} \\ \left(\frac{p_i}{1 + \theta_y} \cdot s_i^e - w \cdot e_i - i_{l,i} \cdot l_{i,-1} + i_d \cdot d_{i,-1}\right) \cdot (1 + \epsilon_{i,\pi}) & \text{if } i \in \{\text{Overconfident}\} \end{cases} \tag{12}$$

$$\epsilon_{i,\pi} \sim \mathcal{N}(\mu_\pi, \epsilon_\pi^2), \ \mu_\pi > 0 \tag{13}$$

$$div_i^e = \max\{(1 - s_f)\pi_i^e, 0\}. \tag{14}$$

Taking inspiration from Fazzari and Mott [70], C-firms have a constant threshold capacity utilization level ($u^T$) below which they never wish to invest. Above that threshold, firms may want to invest or not with a probability ($prob_i^{inv}$, 15) that depends on their current utilization rate. For firms willing to invest, their desired investment ($i_i^{des}$, 16) is a fraction of the investment

required to reach a stock of capital ($k_i^T$, 17) consistent with the threshold utilization rate:

$$prob_i^{inv} = min(\lambda_{i_{des}}^0 + \lambda_{i_{des}}^1 \cdot u_i, 1) \tag{15}$$

$$i_i^{des} = \begin{cases} \lambda_{i_{des}} \cdot P_k \cdot (k_i^T - k_i) & \text{if } k_i^T > k_i \text{ and } \epsilon_i^{i_{des}} < prob_i^{inv} \\ 0 & \text{else} \end{cases} \tag{16}$$

$$k_i^T = y_i^p \cdot v/u^T, \tag{17}$$

where $P_k$ is the price of investment goods and $\epsilon_i^{i_{des}}$ is the realization of a uniform distribution over the interval [0,1].

## 3.3. Capital structure

C-firms wish to hold deposits that serve as a precautionary cushion, following Caiani *et al.* [45]. Each firm has a target range of precautionary deposit levels whose boundaries ($d_{i,min}^T$, 18, and $d_{i,max}^T$, 19) are defined by a fraction of their current expenditures, wage bill and net interest payments. Firms ensure their expected deposits at the end of the period remain within these bounds. Depending on their minimum target deposits and expectations, firms define their expected available internal funds to finance investment ($d_{a,i}^e$, 20):

$$d_{i,min}^T = \lambda_{d,min} \cdot (w \cdot e_i + i_{l,i} \cdot l_{i,-1} - i_d \cdot d_{i,-1}) \tag{18}$$

$$d_{i,max}^T = \lambda_{d,max} \cdot (w \cdot e_i + i_{l,i} \cdot l_{i,-1} - i_d \cdot d_{i,-1}) \tag{19}$$

$$d_{a,i}^e = max(d_{i,-1} + s_f \cdot \pi_i^e - d_{i,min}^T, 0). \tag{20}$$

In order to represent the pecking order theory [71], we assume that if a firm's expected internal funds are insufficient to finance its desired investment, it might decide to raise external finance, either debt or equity, or to decrease its investment expenditures, depending on the relative cost of external financing and on its expected profit rate. To do so, C-firms compute their expected cost of equity ($c_{eq,i}^e$, 21) and debt ($c_{l,i}^e$, 22). Through an arbitrage process ($arb_i$ and $\chi_{arb_i}$, 23 and 24), firms set their preference for equity and debt and deduce from it their expected average cost of external financing ($c_{k,i}^e$, 25):

$$c_{Eq,i}^e = \frac{div_{i,}^e}{p_{Eq_i,-1} \cdot Eq_{i,-1}} \tag{21}$$

$$c_{l,i}^e = i_{l,i} \tag{22}$$

$$arb_i = \frac{c_{l,i}^e - c_{Eq,i}^e}{c_{l,i}^e} \tag{23}$$

$$\chi_{arb_i} = \frac{1}{1 + e^{-\gamma_{arb} \cdot arb_i}} \tag{24}$$

$$c_{k,i}^e = \chi_{arb_i} \cdot c_{l,i}^e + (1 - \chi_{arb_i}) \cdot c_{Eq,i}^e. \tag{25}$$

Firms that lack liquidity might decide to raise external funds with a probability ($prob_i^{ext}$, 26) that depends on the expected cost of external financing relative to the expected profit rate. Otherwise, firms will finance their realized investment ($i_i^{rea}$, 27) out of internal funds only. Capital then accumulates according to the realized investment and a depreciation rate ($\delta$).

$$prob_i^{ext} = min\left(\lambda_{ext}^0 + \lambda_{ext}^1 \cdot \frac{\pi_i^e/(P_k \cdot (k_i + i_i^{des}))}{c_{k,i}^e}, 1\right) \tag{26}$$

$$i_i^{rea} = \begin{cases} i_i^{des} & \text{if } \epsilon_i^{ext} < prob_i^{ext} \\ min(i_i^{des}, d_{a,i}^e) & \text{else} \end{cases} \tag{27}$$

C-firms compute their external financing needs ($rf_i^e$, 28) based on their target and expected deposit levels. Two points are worth mentioning here. First, a firm's financing needs can be negative if its expected deposit level is above the maximum desired one, in which case the firm will use the excess funds to pay back debt and buy back equity. Second, a firm that decided not to raise external funds to finance investments can still have positive financing needs to restore its liquidity level.

$$rf_i^e = \begin{cases} 0 & \text{if } d_{i,min}^T < d_i^{e,int} < d_{i,max}^T \\ \dfrac{d_{i,min}^T + d_{i,max}^T}{2} - d_i^e & \text{else} \end{cases} \tag{28}$$

$$d_i^{e,int} = d_{i,-1} + s_f \cdot \pi_i^e - i_i^{rea}, \tag{29}$$

where ($d_i^{e,int}$, 29) are firms' expected remaining deposits at the end of the period using internal resources only.

C-firms' increased net debt ($\Delta l_i$, 30) and equity emissions ($\Delta Eq_i$, 31) depend on two things. First, firms will raise or decrease their total external liabilities to adjust their deposit level to the target one and to finance investment expenditures. Second, firms will buy back (issue) equity to contract (reimburse) loans while keeping their total external liabilities constant to adjust their current liability structure towards the desired one:

$$\Delta l_i = \chi_{arb_i} \cdot rf_i + \lambda_{arb} \cdot (\chi_{arb_i} \cdot ext_{i,-1}^L - l_{i,-1}) \tag{30}$$

$$\Delta Eq_i = \left(1 - \chi_{arb_i}\right) \cdot \frac{rf_i}{p_{Eq_i,-1}} + \lambda_{arb} \cdot \left((1 - \chi_{arb_i}) \cdot \frac{ext_{i,-1}^L}{p_{Eq_i,-1}} - Eq_{i,-1}\right), \tag{31}$$

where $ext_i^L$ is the firm's total external liabilities (debt + equity), $\chi_{arb_i} \cdot ext_{i,-1}^L$ is the desired fraction of debt in total external liabilities and $(1 - \chi_{arb_i})ext_{i,-1}^L$ is the desired fraction of equity in total external liabilities. C-firms cannot have negative debt or equity, and we do not allow firms to buy back or emit more than 25% of existing equity stock. They will accumulate deposits above their desired maximum level if they have no external liabilities to reimburse.

## 3.4. Demand distribution

Aggregate demand for consumption goods ($Y^D$, 32) is the sum of households' and government consumption expenditures,

$$Y^D = C_g + C_h. \tag{32}$$

We follow [72, 73] and [60] and assume that demand per firm ($ydn_i$, 34, for nominal demand and $yd_i$, 35, for real demand) is distributed between firms using two indicators, firms' relative prices ($\tilde{p}$, 35) and relative capital stocks ($\tilde{k}$, 36). Prices are used as a measure of firms' competitiveness. The capital stock captures the idea that bigger firms attract more demand, irrespective of their price. Firms' share of nominal aggregate demand ($f_i$,37) follows a linear adjustment towards the target share ($f_i^T$, 38). As in Reissl [60] the desired share of aggregate demand is normalized, multiplied by a normally distributed random shock of mean 1 and standard deviation $\sigma_{f_{eq}}$ and then normalized once more to ensure the condition $sum_j f_i^T = 1$ holds.

$$ydn_i = f_i \cdot Y^D \tag{33}$$

$$yd_i = f_i \cdot \frac{Y^D}{p_i} \tag{34}$$

$$\tilde{p} = \frac{p - \text{mean}(p)}{\text{mean}(p)} \tag{35}$$

$$\tilde{k} = \frac{K - \text{mean}(k)}{\text{mean}(k)} \tag{36}$$

$$f_i = \lambda_f \cdot f_i^T + (1 - \lambda_f) \cdot f_{i,-1} \tag{37}$$

$$f_i^T = \lambda_f^p \cdot f_p(\tilde{p}) + (1 - \lambda_f^p) \cdot f_k(\tilde{k}), \tag{38}$$

where $f_p$ and $f_k$ are generalized logistic functions of the form:

$$f_x = \frac{2}{(1 + e^{\gamma_x \cdot \tilde{x}})^{1/v_x}} - 1. \tag{39}$$

This functional form has the advantage of taking values bounded between −1 and +1, with an asymmetric distribution within this interval.

In case firms cannot satisfy demand, which might happen if their current output plus inventory stock is lower than the demand for consumption goods they receive, the unfilled demand is relocated towards other firms. Firms realized profits ($pi_i$, 40), inventories ($inv_i$, 41), deposits ($d_i$, 42) and net worth ($nw_i$, 43) are given by:

$$\pi_i = y_i^d - w \cdot e_i - i_{l,i,-1} \cdot l_{i,-1} + i_d \cdot d_{i,-1} \tag{40}$$

$$inv_i = inv_{i,-1} + y_i^p - y_i^d \tag{41}$$

$$d_i = d_{i,-1} + \pi_i + \Delta l_i + p_{Eq_i,-1} \cdot \Delta Eq_i - I_i - div_i \tag{42}$$

$$nw_i = k_i + inv_i + d_i - l_i. \tag{43}$$

### 3.5. Firms' default

There are two situations that might cause a firm to go bankrupt—illiquidity and insolvency. Firms always ensure they keep a minimum amount of desired deposits so that they run out of liquidity only because of errors in anticipating demand and profits. Firms become insolvent when profits turn negative for a prolonged period of time. When a firm defaults, its capital and inventories are sold to households with a discount $\iota$ to repay the firm's liabilities. If the funds recovered ($recf_i$, 44) are higher than the firm's loans, households will fully bail-in the firm. Otherwise, the banking sector will bear a part of the bail-in cost out of its own profits (45).

$$recf_i = (1 - \iota) \cdot (k_{i,-1} + v_{i,-1}) + d_i \tag{44}$$

$$bd_i = \max(0, l_i - recf_i) \tag{45}$$

Households' net cost associated with the $i^{th}$ firm's default is the difference between what households pay to buy the scrapped capital and inventories of the firm and the part of the recovered funds not used to repay the firm's liabilities to the banking sector:

$$nc_{h,i} = (recf_i - d_i) - (recf_i - (l_i - bd_i)) = l_i - d_i - bd_i. \tag{46}$$

$BD = \sum_i b\,d_i$ represents the aggregate bad debt of banks and $NC_h = \sum_i n\,c_{h,i}$ represents the recovered funds of households.

A defaulted firm is immediately replaced by a new one that receives the capital stock and inventories of the defaulted firm. New firms receive an initial amount of deposits equal to the upper bound of their target deposit level, a fraction of which is raised by issuing debt; the rest is provided by households. The new firm is allocated a random share of aggregate demand. If its productive capacity is insufficient to satisfy this demand during the first period, households will also provide the firms with extra deposits to finance the necessary investments. New firms begin with an initial number of shares $Eq_i = Eq_0$, and an initial share price $p_{Eq_i} = \text{mean}(q) \cdot (k_0 + d_i + n_i)/Eq_i$, where $\text{mean}(q)$ is the average Tobin's Q among non-defaulted firms.

Finally, the new firm has a probability $\epsilon_{\text{overconfident}}$ of being the overconfident type, which will impact its profit expectations through Eq (12). The total cost associated with firms' default and financing of new firms ($TC_h$, 47) and the net financial cost ($NFC$, 48) associated with firms' default are defined by:

$$TC_h = \sum_{i \in \{\text{defaulted}\}} (nc_{h,i} + d_i^{new}) \tag{47}$$

$$NFC = \sum_{i \in \{\text{defaulted}\}} (p_{Eq_i,-1} \cdot Eq_{i,-1} - (recf_i - l_i + bd_i)), \tag{48}$$

where $d_i^{new}$ is total deposits provided to the new firm by households.

## 4. Rest of the economy

As previously stated, the rest of the economy consists of aggregated sectors.

### 4.1. Capital producers

The K-firms sector produces investment goods ($I$, 49) for C-firms using labour ($E_k$, 50) as its only input. We assume labour productivity of K-firms ($A_k$, 51) is a multiple of productivity in the consumption sector. The price of capital goods ($P_k$, 52) is set at the beginning of the period as a constant mark up $mu_k$ over unit costs ($uc_k$,53) and government taxes ($\theta_k$). We assume an

equilibrium capital goods market where output is demand-determined. To keep the model simple, we assume that K-firms redistribute all their profits ($\Pi_k$, 54) to households as dividends $Div_k$ so that their net worth is constantly zero.

$$I = \sum_i (i_i^{rea}) \tag{49}$$

$$E_k = \frac{I}{P_k \cdot A_k} \tag{50}$$

$$A_k = mult_A.A \tag{51}$$

$$P_k = (1 + \mu_k) \cdot (1 + \theta_k) \cdot uc_k \tag{52}$$

$$uc_k = w/A_k \tag{53}$$

$$\Pi_k = Div_k = \frac{I}{1 + \theta_k} - w \cdot E_k. \tag{54}$$

### 4.2. Households

Total employment ($E$, 55) is the sum of public employment and private employment in the capital and consumption goods sectors. The wage bill ($WB$, 56) is given by the wage rate times total employment. Finally, we assume a Phillips curve defined in terms of nominal wages ($w$, 57) in line with the seminal paper by [74], where wage negotiations are nominal and depend positively on employment rate and current inflation, with a degree of money illusion measured by $\omega_2$:

$$E = E_c + E_k + E_g \tag{55}$$

$$WB = w \cdot E \tag{56}$$

$$w = w_{-1} \cdot \left(1 + \omega_0 + \omega_1 \cdot \frac{E}{POP} + \omega_2 \cdot INFL_{-1}\right). \tag{57}$$

Households' total income ($Yh$, 58), net of government taxes, is the sum of their net wage income ($Yh_w$, 59) and their net financial income ($Yh_f$, 60). Households' financial income is itself composed of interest received for their holdings of deposits ($D_h$), government bonds ($B_h$) and dividends received from firms and banks. Households' consumption ($C_h$, 61) follows a linear adjustment towards a target ($C_h^T$, 62) based on their wages in the previous period, financial incomes and net worth; see Godley and Lavoie [64] for a discussion of that functional form. Households' savings ($S_h$, 63) is the difference between their savings and their expenditures:

$$Yh = Yh_c + Yh_c \tag{58}$$

$$Yh_w = (1 - \theta_w) \cdot WB \tag{59}$$

$$Yh_f = (1 - \theta_\pi) \cdot (i_d \cdot D_{h,-1} + i_b \cdot B_{h,-1} + Div_f + Div_k + Div_b) \tag{60}$$

$$C_h = \lambda_{c_h}.C_h^T + (1 - \lambda_{c_h}).C_{h,-1} \tag{61}$$

$$C_h^T = c_0 \cdot Yh_{w,-1} + c_1 \cdot Yh_{f,-1} + c_2 \cdot (D_{h,-1} + B_{h,-1} + MK_{-1}) \tag{62}$$

$$S_h = Y_h - C_h. \tag{63}$$

Households buy all shares emitted by C-firms and all bonds supplied by the government in excess of banks demand for bonds. They suffer losses due to firms' defaults and financing of newly created firms ($TC_h$), they recover funds by scrapping capital of defaulted firms ($RF_h$) and keep their remaining savings as deposits:

$$\Delta D_h = S_h - \sum_{i=1}^{N_f} (p_{E_i,-1} \cdot \Delta Eq_i) - (\Delta B - \Delta B_b) - (TC_h - RF_h) \tag{64}$$

Households' return on equity (ROE) is the sum of dividends and real capital gains per nominal unit of equity, computed for each C-firm ($\rho_i$, 65). Households compute the expected ROE ($\rho_i^e$, 66) using an average of past ROEs. Their expected ROE at the market level ($\rho^e$, 67) is the sum of all firms' expected ROE (weighted by their total equity value) minus the average net financial cost of firms' default $NFC_h$:

$$\rho_i = \frac{div_i}{Eq_i \cdot p_{E_i,-1}} + \frac{\Delta p_{E_i,-1}}{p_{E_i,-1}} - \frac{\Delta p}{p} \tag{65}$$

$$\rho_i^e = \overline{\rho}_i \tag{66}$$

$$\rho^e = \sum_i \rho_i^e \cdot \frac{Eq_i \cdot p_{E_i,-1}}{\sum_i Eq_i \cdot p_{E_i,-1}} - \frac{\overline{NFC_h}}{MK}, \tag{67}$$

where $\overline{x}$ is the average value of variable $x$ for the four previous periods.

We posit that households then compute a target equity to non-risky assets (i.e. deposits and government bounds) ratio ($\sigma^T$, 68) that depends on the spread between the expected ROE at the market level and the average return on safe assets ($\tilde{i}$, 69). Households adjust their total equity holding ($EQ_c$, 70) using an actual share ($\sigma$, 71) converging with speed $\lambda_\sigma$ to this target:

$$\sigma^T = \sigma_0^T + \sigma_1^T \cdot (\rho^e - \tilde{i}) \tag{68}$$

$$\tilde{i} = (i_d \cdot D_{h,-1} + i_b \cdot B_{h,-1})/(B_{h,-1} + D_{h,-1}) \tag{69}$$

$$EQ_c = \sigma(D_{h,-1} + B_{h,-1}) \tag{70}$$

$$\sigma = \frac{EQ_c}{D_{h,-1} + B_{h,-1}} = \lambda_\sigma \cdot \sigma_{-1} + (1 - \lambda_\sigma) \cdot \sigma^T. \tag{71}$$

The resulting aggregate equity value is then allocated between the different C-firms depending on their relative individual performances. We assume there are both chartists and fundamentalists within the households sector so that the target share of total market capitalization to allocate to each firm ($f_{Eq,i}^T$, 72) depends on the firm's past average financial returns and its Tobin's Q ($q_i$, 73), both relative to other firms ($\tilde{q}_i$ and $\tilde{\rho}_i$, 74 and 75). Households' realized portfolio allocation ($f_{Eq,i}$, 76) follows a linear adjustment towards the targeted one. Finally, the

equity price ($p_{eq,i}$, 77) adjusts according to households' realized portfolio allocation:

$$f_{Eq,i}^T = \frac{1}{N_f}(1 + \lambda_{f_{eq}}^{ret} \cdot f_{ret}(\tilde{\rho}_i) + (1 - \lambda_{f_{eq}}^{ret}) \cdot f_{nw}(\tilde{q}_i)) \tag{72}$$

$$q_i = \frac{p_{E_i,-1} \cdot Eq_i}{nw_{i,-1}} \tag{73}$$

$$\tilde{q}_i = \frac{\overline{q}_i - \text{mean}(\overline{q}_i)}{\text{mean}(\overline{q}_i)} \tag{74}$$

$$\tilde{\rho}_i = \frac{\overline{\rho}_i - \text{mean}(\overline{\rho}_i)}{\text{mean}(\overline{\rho}_i)} \tag{75}$$

$$f_{Eq,i} = \lambda_{feq} \cdot f_{Eq,i}^T + (1 - \lambda_{feq}) \cdot f_{Eq,i,-1} \tag{76}$$

$$p_{Eq,i} = \frac{f_{Eq,i} \cdot EQ_c}{Eq_i}, \tag{77}$$

where $\overline{x}$ is again the time-average value of the variable $x$ over the four previous periods and $\tilde{x}_i$ is the spread, in percent, between this time-average for the $i^{th}$ firm and the average value of $\overline{x}$ across all C-firms. Regarding the demand distribution allocation procedure, the vector of target allocations is normalized to ensure $\Sigma f_{Eq,i} = 1$, multiplied by a normally distributed random shock and normalized again. Finally, $f^{ret}$ and $f^{nw}$ are two generalized logistic functions as presented in the demand distribution mechanism.

## 4.3. Banks

The banking sector provides loans ($L$) to firms on demand and banks take firms' and households' deposits ($D_h$ and $D_f$), hold government bonds ($B_b$) and hold reserves ($R_b$). The interest rate the banking sector charges firms ($i_{l,i}$, 78) adjusts towards a target ($i_{l,i}^T$, 79) that depends, following Rzeszutek et al. [75], on three components—the (exogenous) central bank interest rate ($i_{CB}$), a banking sector-specific term ($i_{lb}$, 80) and a firm-specific component that depends on a firm's indebtedness and its size relative to other firms. The banking sector-specific term is updated following the actual level of own funds of banks ($OF_b$) and their targeted level ($OF_b^T$). Deposit holders are remunerated by an interest rate ($i_d$, 81) that depends on inflation:

$$i_{l,i} = \lambda_l i_{l,i}^T + (1 - \lambda_l) i_{l,i,-1} \tag{78}$$

$$i_{l,i}^T = i_{CB} + i_{lb} + \gamma \cdot \frac{l_i}{nw_i + l_i} \cdot \frac{1}{1 + k/max(k)} \tag{79}$$

$$i_{lb} = i_{lb,0} + i_{lb,1} \cdot \frac{OF_{b,-1}}{OF_b^T} \tag{80}$$

$$i_d = \lambda_{id} \cdot (INFL + id_0) + (1 - \lambda_{id}) \cdot i_{d,-1}. \tag{81}$$

As in Yilmaz and Godin [76], banks have a constant target mandatory liquidity ratio ($\zeta_b$) and wish to have enough own funds (82) to satisfy this target. Whenever this is not the case,

banks increase their own funds ([83]) by retaining some of their profits ($Pi_b$, [84]) in the form of retained earnings ($RE_b$, [85]). They distribute a minimum fraction of their profits ($1-s_{b,max}$) to households as dividends ($Div_b$, [86]). Finally, we assume that banks buy a constant share of emitted public bonds ($B_b$, [87]):

$$OF_b^T = \zeta_b.L \qquad (82)$$

$$OF_b = OF_{b,-1} + RE_b \qquad (83)$$

$$\Pi_b = \sum_i i_{l,i,-1}.l_{i,-1} + i_b.B_{b,-1} - i_d.(D_{h,-1} + D_{f,-1}) - BD \qquad (84)$$

$$RE_b = \begin{cases} 0 & \text{if } OF_b^T < OF_{b,-1} \\ \max(s_{b,max} \cdot \Pi_b, OF_b^T - OF_{b,-1}) & \text{if } OF_b^T > OF_{b,-1} \end{cases} \qquad (85)$$

$$DIV_b = \Pi_b - RE_b \qquad (86)$$

$$\Delta B_b = \beta \cdot \Delta B. \qquad (87)$$

## 4.4. Government

We assume that government expenditures ($C_g$, [88]) follow a linear adjustment towards a target ($C_g^T$, [89]) that is a function of unemployment, wages and previous-period nominal GDP (taking into account inflation expectations). Government income consists of taxes ($T_g$, [90]) on consumption and investment firms sales, households' wages and dividends. We further assume the government hires a constant proportion of the population to provide public services ($E_g$, [91]). It finances its excess expenditures by issuing bonds ($B$, [92]) that are sold at a constant price (normalized to one for simplicity) and provides an interest rate ($i_b$, [93]) that depends on the inflation rate:

$$C_g = \lambda_{c_g}.C_g^T + (1 - \lambda_{c_g}).C_{g,-1} \qquad (88)$$

$$C_g^T = G_0 \cdot \left( Y_{-1}^d \cdot (1 + INFL_{-1}) + I_{-1} \cdot \frac{P_k}{P_{k,-1}} \right) + G_1 \cdot w \cdot (POP - E) \qquad (89)$$

$$T_g = \theta_y \cdot Y^d + \theta_k \cdot P_k \cdot I + \theta_{div} \cdot DIV + \theta_w \cdot w \cdot E \qquad (90)$$

$$E_g = \tau_G \cdot POP \qquad (91)$$

$$B = B_{-1} + C_g + w \cdot E_g + i_b B_{-1} - T_g \qquad (92)$$

$$i_b = \lambda_{ib} \cdot (INFL + i_{b,0}) + (1 - \lambda_{ib}) \cdot i_{b,-1}. \qquad (93)$$

## 5. Baseline results

We now turn to the focus of this analysis—understanding the macroeconomic and microeconomic impacts of management practices on the capital structure of firms in Poland. To that end, we run numerical simulations of the model. Each simulation lasts 900 periods and

converges to a quasi-steady state after 500 periods. We therefore show only the last 400 periods, corresponding to 100 years. In each simulation, the economy is composed of 500 C-firms and the four aggregated sectors—households, K-firms, banks and the government. Finally, due to the presence of stochastic elements, we run 100 Monte Carlo (MC) simulations using the baseline calibration of the model from S1 Table. Unless otherwise specified, all the plots presented are averages over the MC simulations.

The model is calibrated for Poland, taking the period 2012Q1–2019Q4 as a reference. We decided not to use a longer time series because Poland has gone through significant structural change over the nearly three decades since the fall of the Berlin wall, integration into the European Union and the great financial crisis of 2008, which had long-lasting effects. The calibration procedure was designed to allow the model to reflect the structure of the economy, both in terms of production structure and financial structure.

In terms of production structure, we calibrated the ratio of consumption,investment and government expenditure to GDP over the period. These time series are stationary over the period considered. We also calibrated the model to the mean level of employment, even though it is not stationary. For the financial structure, we could not directly use the national account because the model greatly simplifies the financial structure of the economy, ignoring interbank lending and stocks such as insurance technical reserves and assuming that all the firms are traded on the financial market. Furthermore, there is a large discrepancy between non-financial transactions (e.g. flows such as GDP, income distribution and interest payments) and financial transactions (flows such as changes in liabilities or assets). We therefore decided to reconstruct financial time series based on non-financial transactions so that they match each other. We calibrated the model to the constructed level of public and private debt. Fig 1 displays the Polish time series of the GDP structure and the simulated one for the baseline quasi steady-state. The other targets used for the calibration were inflation (simulated average: 1.3%, observed average: 1.04%), public debt to GDP (simulated average: 0.78, observed average: 0.77), private debt to GDP (simulated average: 1.06, observed average: 1.14) and unemployment rate (simulated average: 6.2%, observed average: 6.7%). The values for all parameters were displayed in S1 Table.

Finally, we set the proportion of overconfident firms at 60%, using the results of (Rzeszutek and Szyszka 2020).

## 5.1. Validation

After having calibrated the production and financial structure, we conducted a short validation exercise, concentrating on macroeconomic aspects. We follow the strategy described in Assenza, Gatti, and Grazzini [77] and Caiani et al. [45] and compare volatility, auto-correlation and cross-correlation structures of main aggregates with their empirical counterparts. Trends and cyclical components have been separated using the Hodrick–Prescott filter.

Fig 2 display the recurrent components of GDP, investment, unemployment and government expenditure, each normalized by the trend for ease of comparison. We observe, as in real data, that investment, unemployment and government expenditure are more volatile than GDP. For investment, the mean observed value is 5.11, while it is 4.68 in the simulated time series. In the case of unemployment, the mean observed value is 9.02, while it is 8.32 in the simulated time series. In the case of government expenditure, the mean observed value is 1.01, while it is 1.36 in the simulated time series.

Figs 3 and 4 display the auto-correlation and cross-correlation structures of the de-trended time series up to the 20th lag (10th lag for cross-correlation). Both structures are similar to the observed structure with decreasing auto-correlation, even if the simulated time series tend to be less auto-correlated.

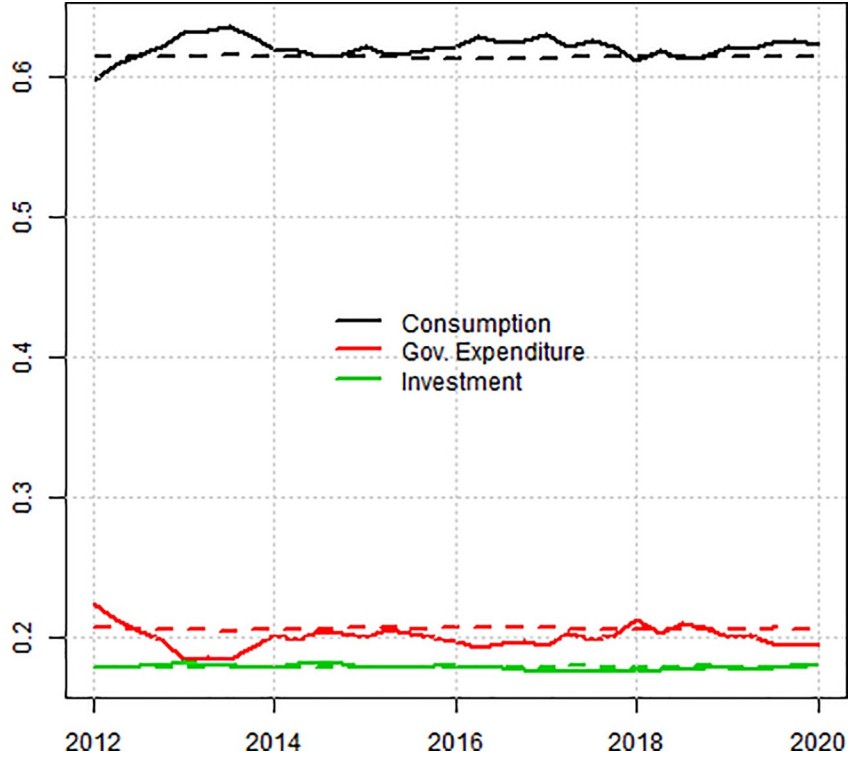

**Fig 1. Simulated (dotted) and Polish (solid) time series used for calibration of the structure of GDP.** Source: OECD quarterly national accounts and authors' computations.

## 5.2. Baseline analysis

We now introduce in the model overconfident firms that tend to overestimate their future profits when formulating expectations. We set the proportion of overconfident firms at 60% using the results of Rzeszutek and Szyszka [63]. Key results of the baseline simulations are presented in Fig 5. All results are averages of 100 MC simulations unless specified otherwise. Finally, all flow variables are averaged over four periods to give annual averages.

As seen in Fig 5, the model correctly replicates the results found in the literature. First, overconfident firms demonstrate a higher investment propensity, as illustrated by their higher investment to output ratio (panel 1). This is because they tend to overestimate their future cash flows (12), hence overestimating both expected future ROI and the amount of internal funds available to finance investments (20). Second, overconfident firms demonstrate a lower net equity emission than neutral firms (panel 5). Because they indeed overestimate their future profits, overconfident managers perceive their company stock as being undervalued, thus considering equity emissions to be excessively costly. This can encourage these firms to realize pure arbitrage operations; they might issue debt, which is perceived as less costly, solely to buy back equity. Third, as overconfident firms rely more on debt than equity emissions, they have a higher debt to output ratio (panel 4). Finally, these firms are more likely to decide to finance investment using only internal funds, as they find external financing too costly (26). In addition, as they overestimate their future cash flows, overconfident firms tend to underestimate their need for external funds to satisfy their liquidity cushion requirements (Eqs 18 and 19). Because of these two effects, overconfident firms have a lower deposit to output ratio than neutral firms (panel 3).

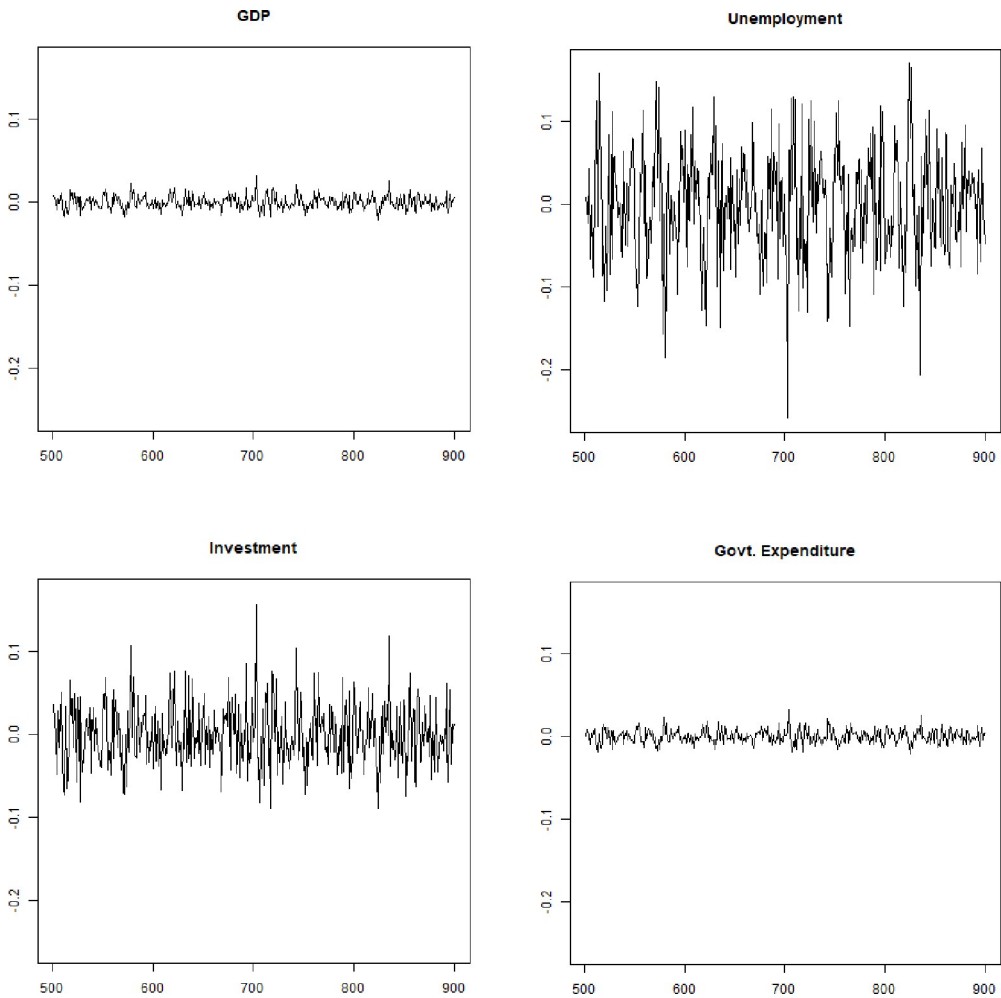

**Fig 2. Cyclical components of simulated times series for GDP, unemployment, investment, and government expenditure.** Source: Authors' computations.

We see that overconfident firms are on average bigger than neutral ones, as they attract a higher share of demand and equity (panels 2 and 6). However, their lower deposits and higher indebtedness indicate these firms tend to take more risks and are thus more likely to default (6.5% vs. 2.5%). On the contrary, by correctly anticipating their profits neutral firms are financially stronger and much less likely to default, but they are also smaller. Firms thus face a trade-off between growth and stability. In addition to their default rate, overconfident firms' risk-taking is illustrated by the larger spread of their quartiles at 25% and 75% (clearly seen for investment, debt, net equity emissions and share of market capitalization), indicating that not all overconfident firms are successful.

## 5.3. Sensitivity analysis

In order to better understand the micro and macroeconomic impact of excess overconfidence, we run a sensitivity analysis on the proportion of overconfident firms in the economy ($prop_{opt}$) and on the average bias in their profit expectations ($\mu_\pi$), i.e. their average degree of overconfidence. The proportion of overconfident firms ranges from 20% to 80%, with an increment of 5%, and the average degree of overconfidence ranges from 2% to 20%, with an increment of

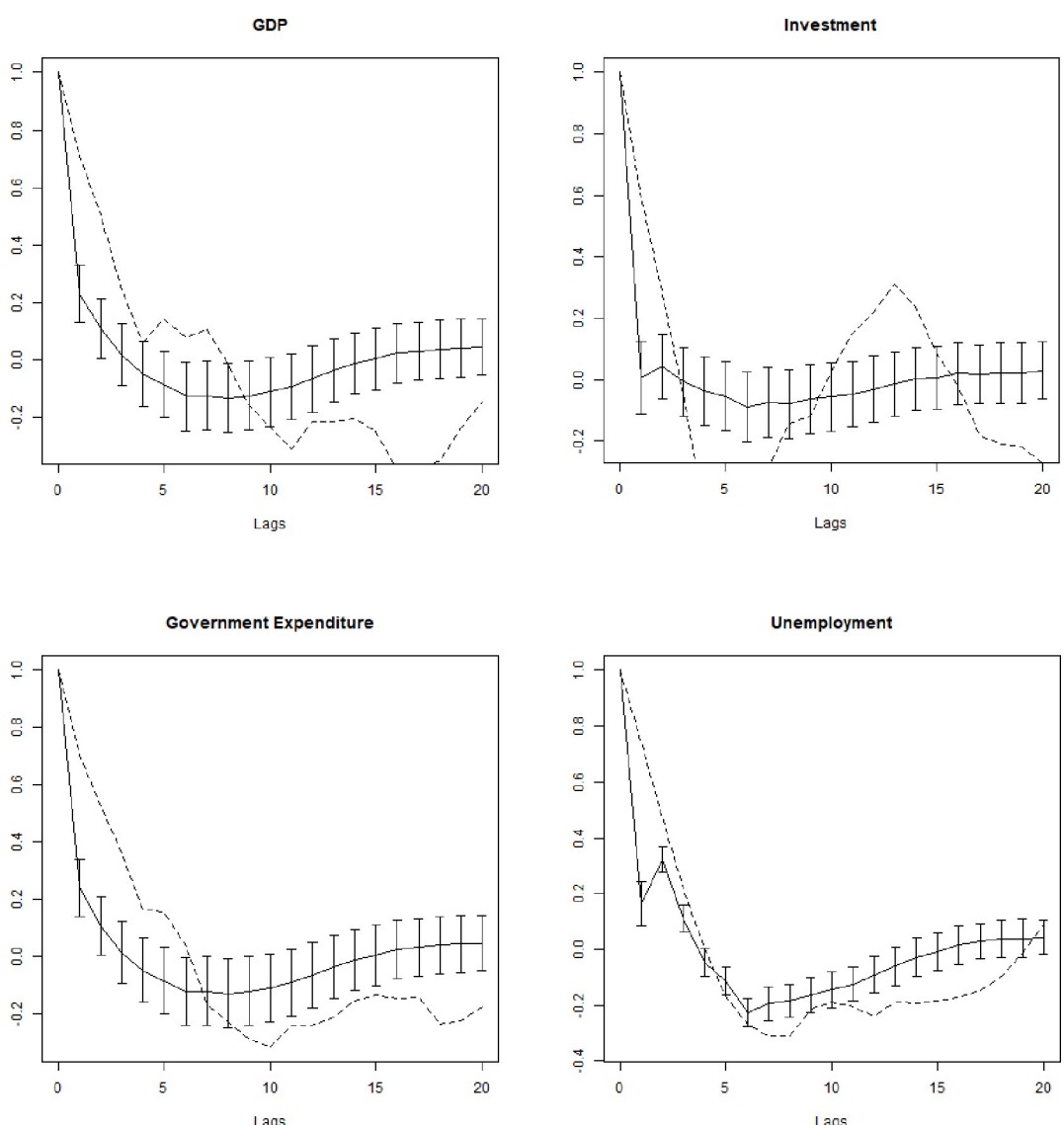

**Fig 3. Average artificial (continuous) and real (dashed) auto-correlations of the de-trended series up to the 20th lag.** Bars are standard deviations of Monte Carlo average auto-correlations.

2%. For each set of values, we run 10 Monte Carlo (MC) simulations and look at the average default rate, debt to output ratio, interest rate and coefficient of variation of output growth. The x and y axes give the percent of overconfident firms and their average degree of overconfidence, respectively. The colour of each cell gives the average value over time (for periods 501 to 900) and over MC simulations for each variable, where blue (respectively yellow) denotes lower (respectively higher) values. Unless otherwise specified, all results are averages in the all economy, and we do not distinguish between overconfident and neutral firms. On Fig 6 we see the GDP growth coefficient of variation and Return on Equity, C-firms probability of default, debt to output ratio and interest rate as a function of the proportion of optimistic firms and their degree of optimism. Blue (respectively yellow) denotes lower (respectively higher) values for the given variables.

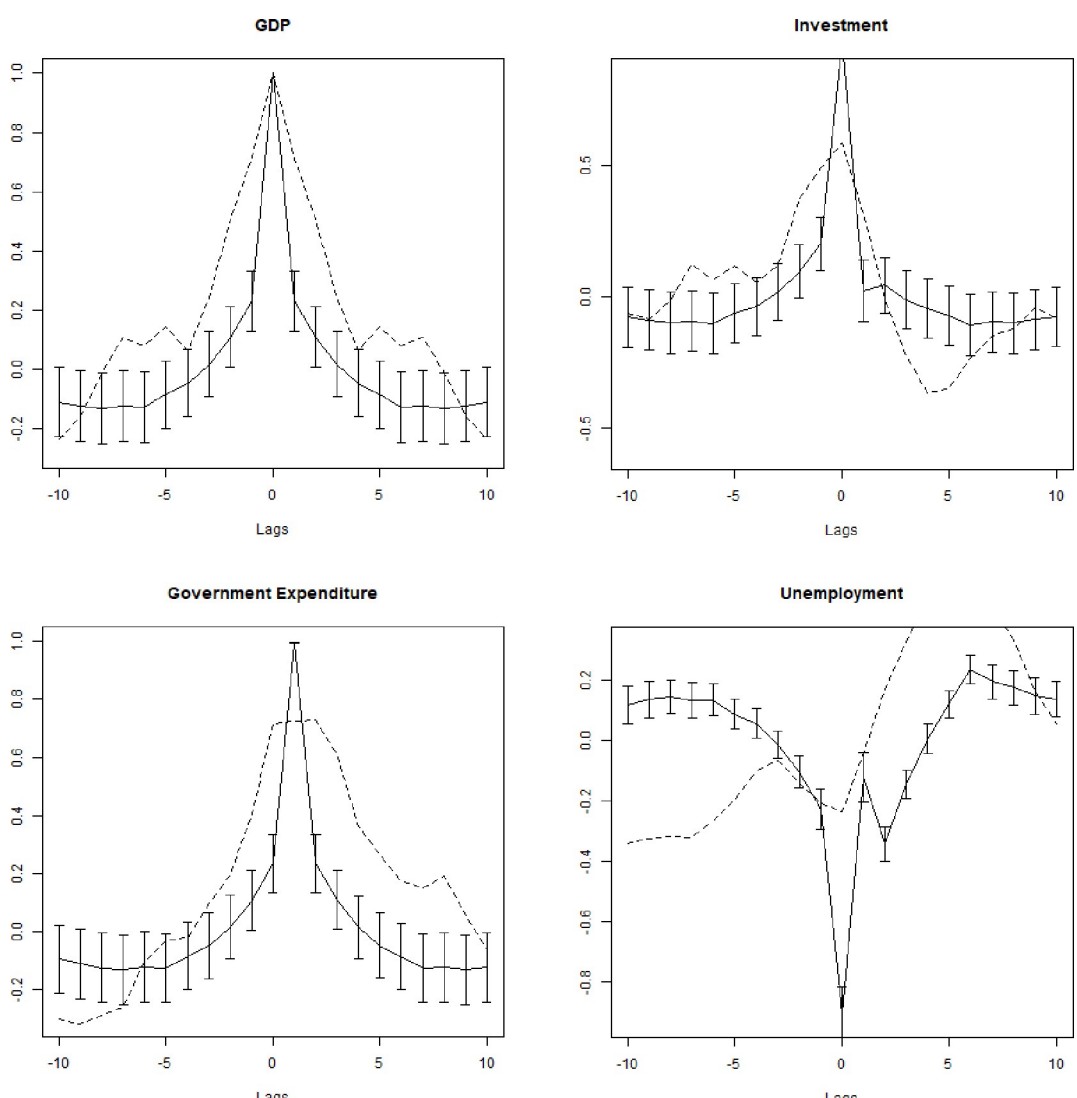

**Fig 4. Average artificial (continuous) and real (dashed) cross-correlations of the de-trended series up to the 10th lag.**
Bars are standard deviations of Monte Carlo average cross-correlations.

We see that increasing either the proportion of overconfident firms or their degree of over-confidence has a destabilizing impact on the economy. Overconfidence affects both the financial sector, as the coefficient of variation of the average return on equity increases (panel 2), and the real economy, as it increases the coefficient of variation of output growth (panel 1). The main reason for this is that higher risk taking by overconfident firms increases the average default rate in the economy (see panel 3). We see that both a higher proportion of overconfident firms and a higher average degree of overconfidence decrease the debt to output ratio (panel 4). At first sight, this seems to contradict the results in the previous section, where we saw that overconfident firms are more indebted than neutral ones on average. However, it would be a fallacy of composition to conclude that the macroeconomic impact of the excess overconfidence of firms' managers can be based on the microeconomic results of the previous section. In fact, we see in panel 6 that the results of the previous section still hold for all sets of parameters; overconfident firms have a higher average debt to output ratio than neutral firms.

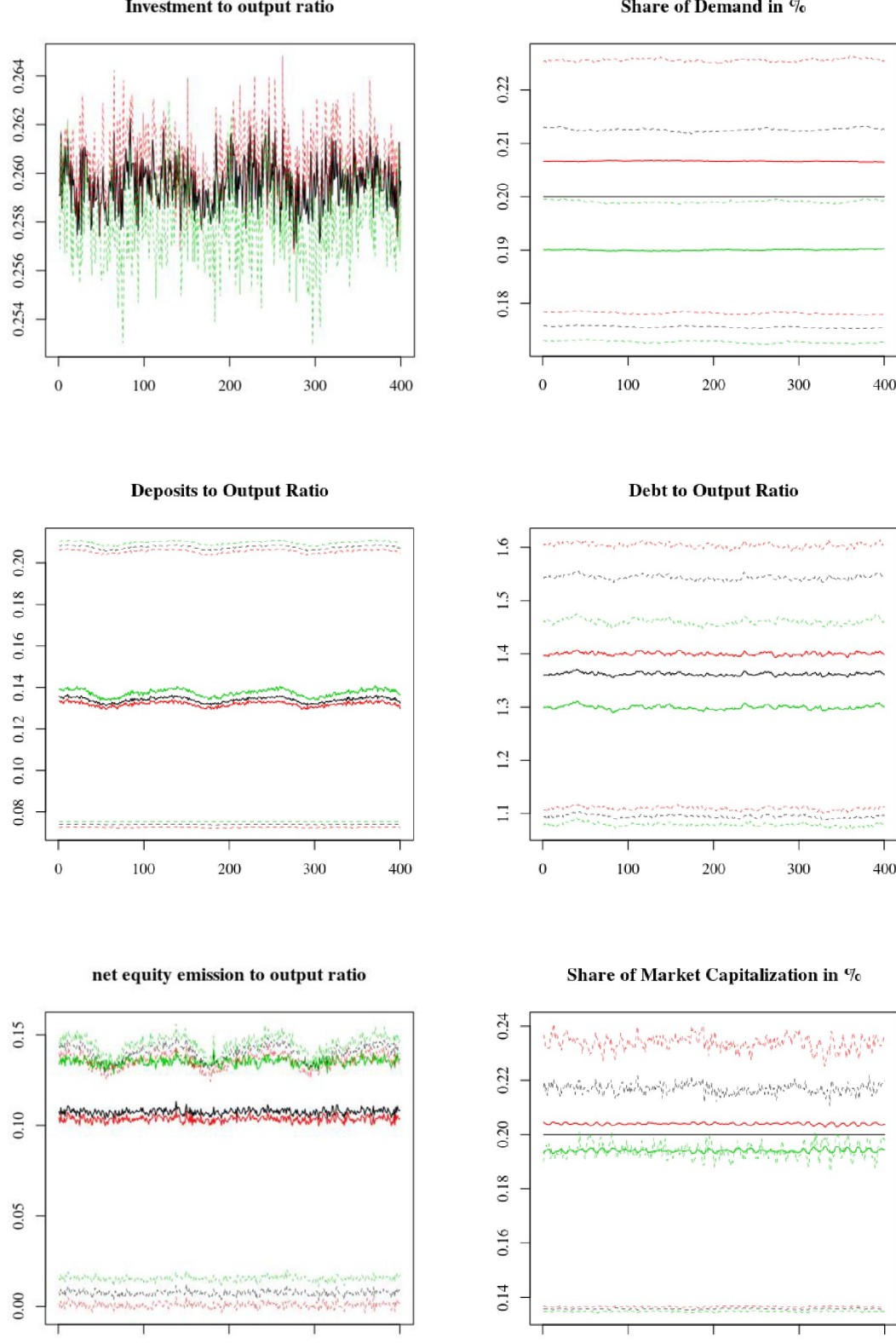

**Fig 5. Selected microeconomic indicators.** Average results, weighted by firms' share of demand (black lines for average across all firms, red lines for overconfident firms and green lines for neutral firms). Means (thick solid lines) and quantiles at 25% and 75% (dotted lines).

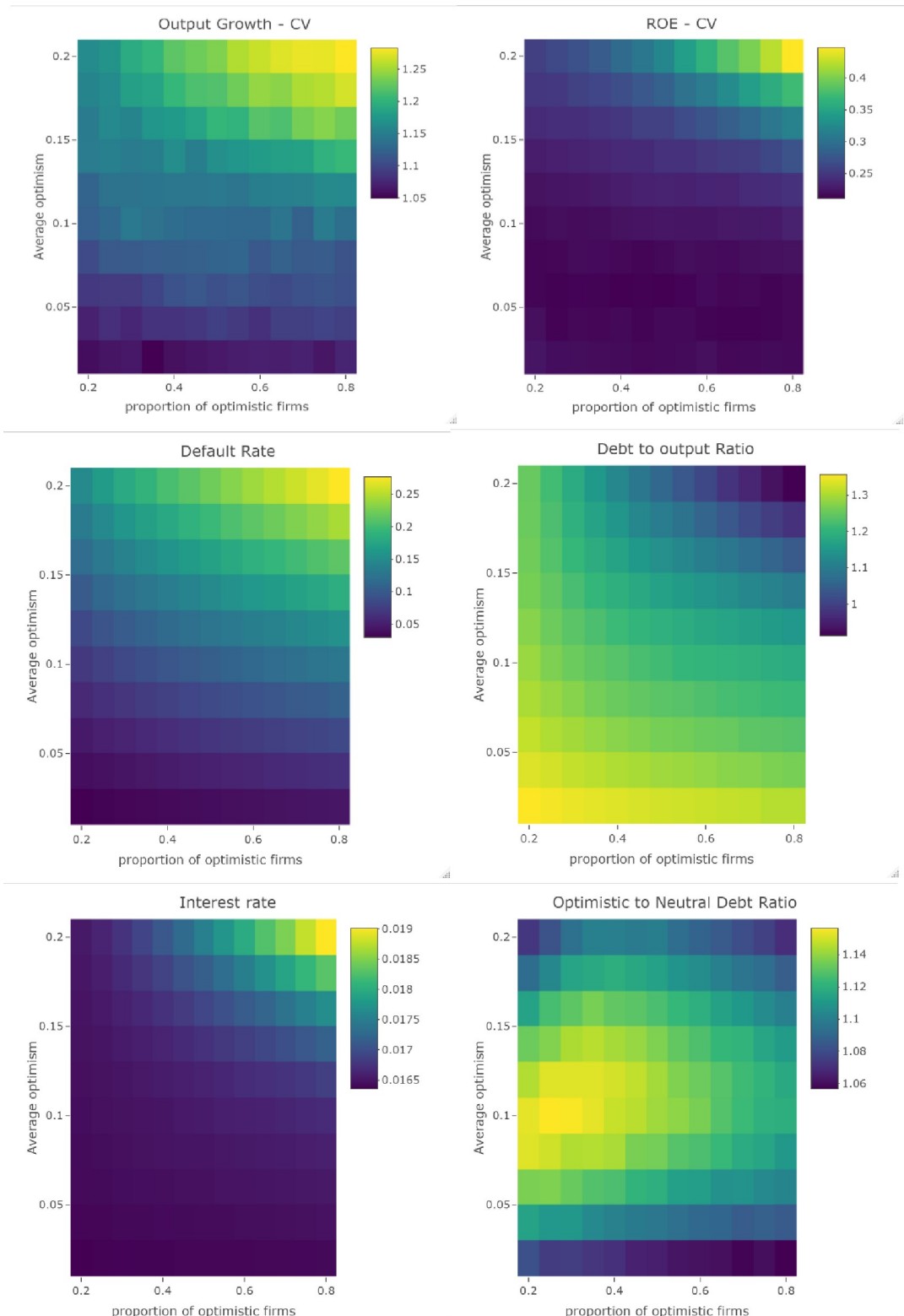

**Fig 6. GDP growth coefficient of variation and return on equity, C-firms probability of default, debt to output ratio and interest rate as a function of the proportion of overconfident firms (X axis) and their degree of overconfidence (Y axis).** Blue (respectivelly yellow) denotes lower (respectively higher) values for the given variables.

The mechanism explaining this result is that the higher default rate associated with excess overconfidence increases losses of the banking sector, as more firms fail to repay their debt. To preserve their capital adequacy ratio, banks increase the average interest rate; see panel 5. This higher interest rate affects overconfident and neutral firms equally, though (80). Facing a higher cost of debt, all firms arbitrage in favour of equity rather than loans (23) or decide to use more internal funds (26), thus reducing their indebtedness, even if overconfident firms still have a higher debt ratio than neutral ones. Finally, we see that an increase in the degree of overconfidence has a greater impact on the economy than an increase in the proportion of overconfident firms. For instance, starting from $mu_\pi$ = 0.1 and $prop_{opt}$ = 0.4, where the default rate is 9.5%, doubling the degree of overconfidence more than doubles the default rate, which increases to 22%, while doubling the proportion of overconfident firms while keeping the degree of overconfidence constant would increase the default rate by roughly 30% to 12.38%.

## 6. Policy shocks

To understand how the economy presented in the previous section reacts to different shocks, we run three different scenarios—a run to liquidity by investors, a monetary policy tightening by the central bank and a decrease in the liquidity cushion by firms. All scenarios consist of a shock in one or more parameters at the time period 600. Again, we present average results over 100 MC simulations and plot only 100 years of simulation (i.e. 400 periods) after an initialization phase of 500 periods.

### 6.1. Run to liquidity

For the first shock, we simulate an increase in risk aversion by households that decide to reallocate their portfolio away from equity and towards safer assets, such as deposits and government bonds. This is represented by a shock in the constant term $\sigma_0^T$ in the target safe assets to equity ratio of households (68). At $t$ = 600, the parameter shifts from $\sigma_0^T = 1.3$ to $\sigma_0^T = 0.9$. The results are presented in Fig 7 and Table 3.

We see that the liquidity shock leads to a reallocation of firms' liability structure. As it becomes more expensive to raise equity, firms increase their debt holding (panel 4) and decrease their equity (panel 5). Du to firms' higher leverage ratio, the interest rate on loans increases moderately for all firms (panel 6). We also see that the higher cost of external liabilities, both equity and loans, encourages firms to decrease their investment, leading to a higher average utilization rate (panel 3). The economy nonetheless manages to adapt to the change in households' risk aversion. Indeed, there is only a very moderate increase in firms' default rate (panel 1). However, we note that the economy goes through a period of high volatility following the shock, with dampened oscillations lasting for roughly 50 periods before stabilizing on a new stationary state. In response to the shock on $\sigma_0^T$, and everything else being equal, households would have decreased their equity holding by 24%. However, we see that the shock caused a much lower decrease in equity holding of only 9.5%. There are two reasons for this. The first is that part of the adjustment occurred through prices rather than quantity; a higher dividend per unit of equity allowed mitigating the run to safety of households. The second is that as firms increase their indebtedness to compensate for the lower available equity financing, household deposits increase. Therefore, households hold more safe assets to use as collateral to purchase equity (71).

To look more closely at the difference between overconfident and neutral firms, we show in Table 3 the average change in the steady state value of the variables displayed in Fig 7 before and after the shock. More precisely, we show the difference, in percentage, between the average value for the period 500 to 600 and the period 800 to 900, both taken as measures of the two

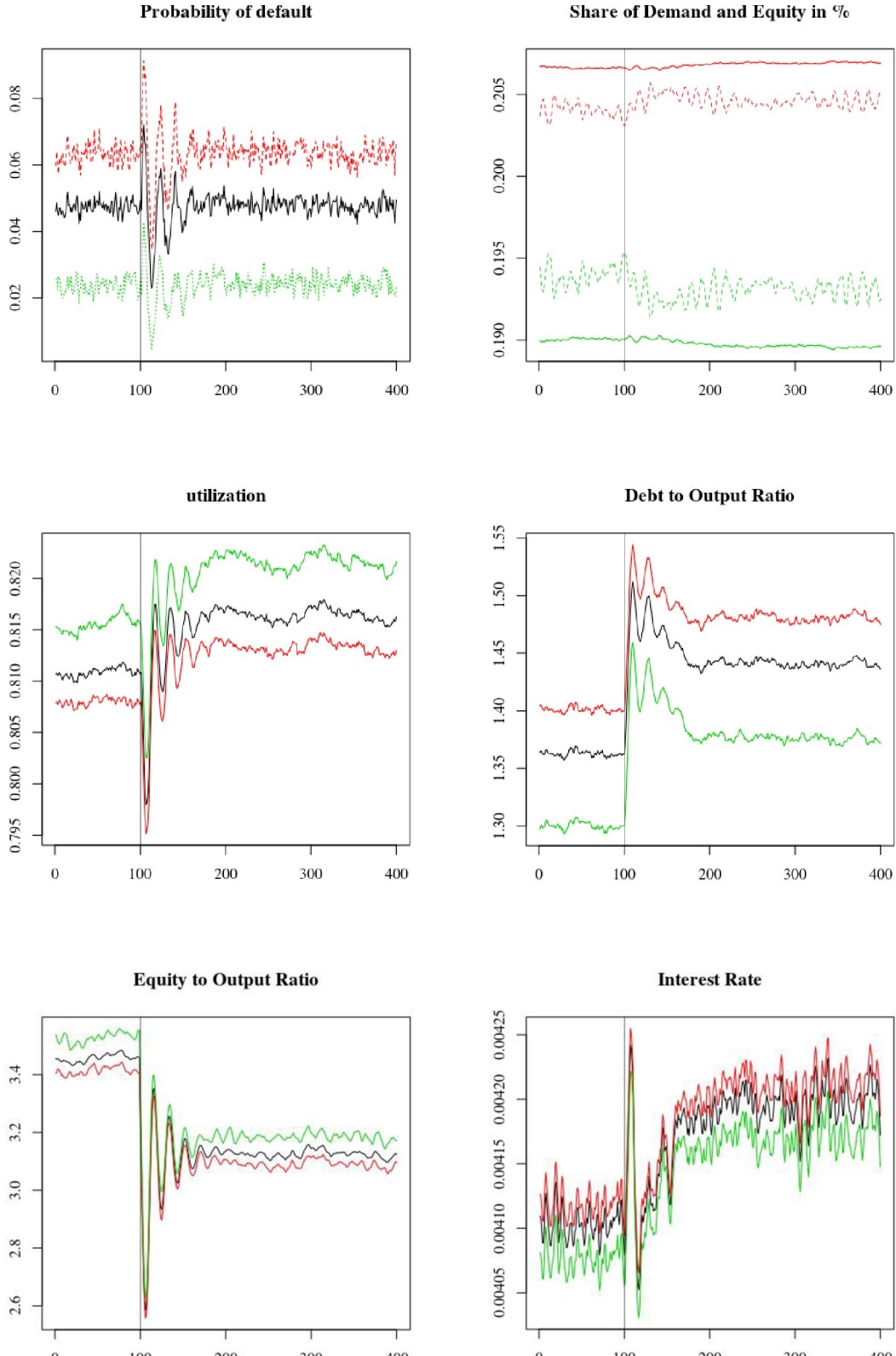

**Fig 7. Run to liquidity: Increase in the parameter $\sigma_0^T$ at $t$ = 600.** Average firms (black), overconfident firms (red) and neutral firms (green).

**Table 3. Shock on parameter $\sigma_0^T$.**

|  | Default | Demand | Mkt. Cap. | Utilization | Debt | Equity | $i_l$ |
|:---:|:---:|:---:|:---:|:---:|:---:|:---:|:---:|
| Average | 1.51 | 0 | 0 | 0.69 | 5.73 | -9.51 | 2.31 |
|  | (6.76) |  |  | (0.29) | (0.25) | (0.07) | (0.30) |
| Overconfident | 1.16 | 0.19 | 0.30 | 0.66 | 5.67 | -9.38 | 2.29 |
|  | (9.92) | (2.77) | (3.13) | (0.39) | (0.30) | (0.12) | (0.33) |
| Neutral | 4.62 | -0.31 | -0.45 | 0.75 | 5.82 | -9.70 | 2.33 |
|  | (4.38) | (2.81) | (3.24) | (0.44) | (0.32) | (0.16) | (0.32) |

Average variation of default probability (Default), share of demand (Demand), share of total market capitalization (Mkt. Cap.), utilization, debt to output ratio (Debt), equity to output ratio (Equity) and interest rate ($i_l$) between steady state before and after the shock, in percentage. Coefficients of variation are in parentheses.

steady state values. We see that overconfident firms are less affected by the shock than neutral firms, as the change in the value of all variables in Table 3 is higher for neutral firms than for overconfident ones. In addition, we see that the changes in the share of total market capitalization and total demand are both of opposite signs between overconfident and neutral firms; overconfident firms benefit from the shock and manage to attract a higher share of equity and demand. This is because, as explained earlier, overconfident firms rely less on equity than neutral ones. This lower use of equity funds is due to both their lower use of external funds, and, when issuing external liabilities, to their preference for liability rather than equity. In any case, as they are less dependent on equity financing, overconfident firms are less affected by this adverse shock and manage to gain equity and demand shares relative to neutral firms following the shock.

The conclusion of this first policy experiment is that we can observe a change in capital structure and investment decisions by firms coming out of a change in investors' behaviour, indicating the importance of correctly accounting for the interdependence between the different sectors of the economy. Furthermore, we see that the behaviour of managers has significant impact on the performance of firms, with more overconfident firms being less dependent on the equity market, attracting more demand and growing faster.

## 6.2. Monetary policy tightening

We now turn to simulating a tightening of monetary policy by the central bank, which increases the cost of debt financing for firms. More precisely, we increase the (annual) central bank interest rate from $i_{cb}$ = 0.6% to 1.2% at $t$ = 600. The results are presented in Fig 8 and Table 4. Again, we plot averages over 100 MC simulations from period 500 to period 900.

Through (23), firms react to the tightening of banking credit by arbitraging more in favour of equity than debt. We see that firms significantly decrease their indebtedness following the shock (panel 4). However, there is no increase in equity, and the equity to output ratio even slightly decreases (panel 5). This is because firms deleveraging decreases the amount of deposits that households could use as safe assets to hedge the risks associated with equity. This reduces their demand for equity, which more than compensates for the arbitrage operations of firms in favour of equity rather than debt. Similar to the previous shock, the tighter financing conditions have a negative impact on investment, leading to a higher utilization rate (panel 3). By limiting firms' access to both debt and equity, the tighter credit conditions have a negative impact on firms' financial soundness and cause an increase in firms' default (panel 1). Finally, we note that an increase of 60 basis point in the central bank interest rate leads to an increase in the central bank interest rate of only 40 basis point, indicating partial pass-through. Similar

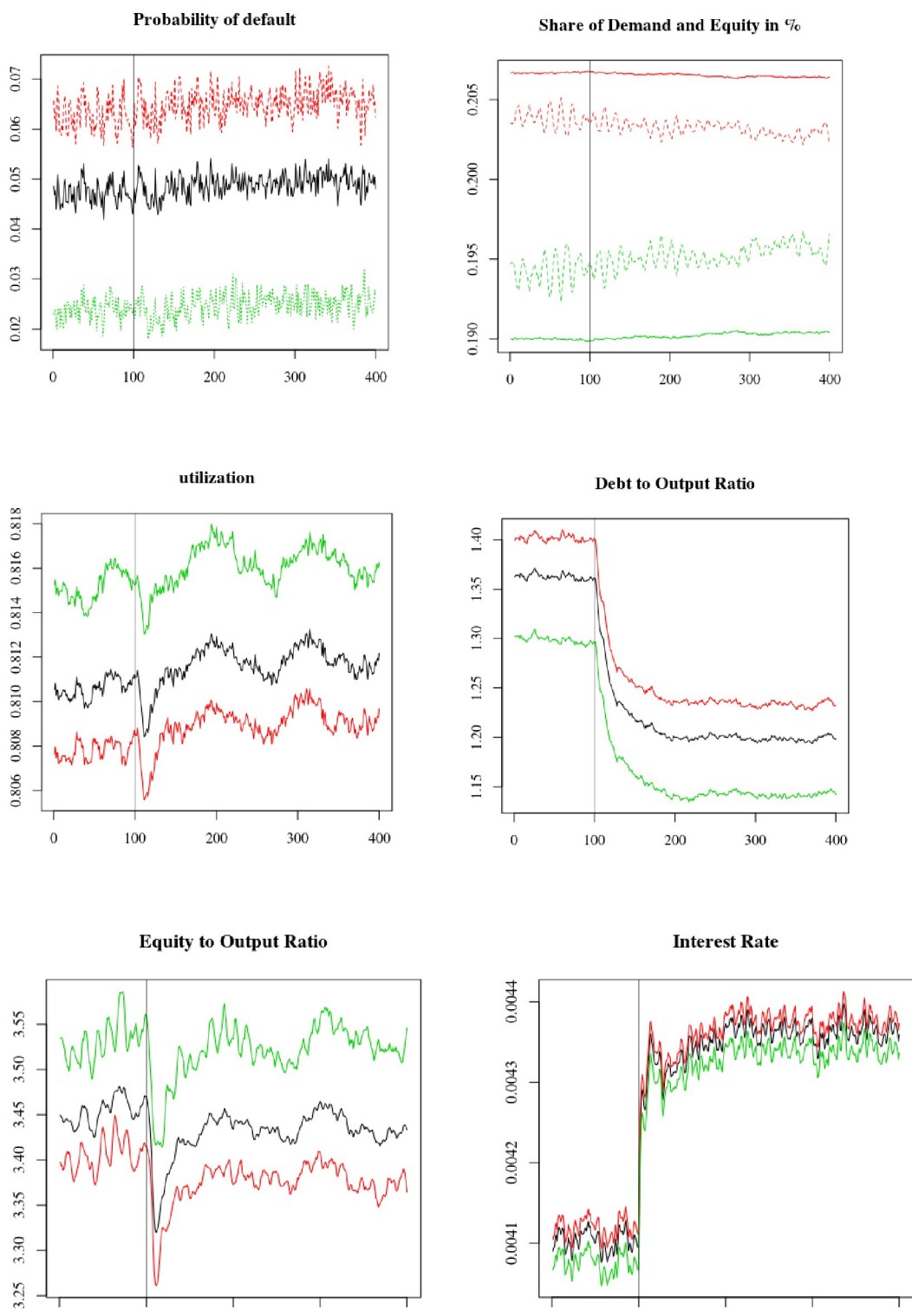

**Fig 8. Monetary policy tightening: Increase in the parameter $i_{cb}$ at t = 600.** Average firms (black), overconfident firms (red) and neutral firms (green).

**Table 4. Shock on parameter $i_{cb}$.**

|  | Default | Demand | Mkt. Cap. | Utilization | Debt | Equity | $i_l$ |
|---|---|---|---|---|---|---|---|
| Average | 6.36 | 0 | 0 | 0.17 | -12.1 | -0.46 | 6.32 |
|  | (1.69) |  |  | (1.14) | (0.09) | (1.66) | (0.1) |
| Overconfident | 6.32 | -0.15 | -0.48 | 0.19 | -12.1 | -0.82 | 6.25 |
|  | (1.98) | (3.30) | (1.96) | (2.40) | (0.12) | (1.45) | (0.1) |
| Neutral | 8.36 | 0.24 | 0.77 | 0.14 | -12.1 | 0.1 | 6.45 |
|  | (2.66) | (3.26) | (1.93) | (2.1) | (0.11) | (16.22) | (0.11) |

Average variation of default probability (Default), share of demand (Demand), share of total market capitalization (Mkt. Cap.), utilization, debt to output ratio (Debt), equity to output ratio (Equity) and interest rate ($i_l$) between steady state before and after the shock, in %. Coefficients of variation are in parentheses.

to the previous policy shock, this is due to the adjustments in firms' external liability structure and the change in households' deposits that affect firms' debt ratio.

We now compare how overconfident and neutral firms react to the shock. We see in Table 4 that overconfident firms are most affected by the shock, as they have to realize higher adjustments than neutral firms for all variables observed except debt. Moreover, contrary to the previous shock, it is overconfident and not neutral firms that lose shares of demand and equity. This result is the exact opposite to what we observed in the previous section where the adverse shock was on equity. Here, as overconfident firms have a higher debt ratio than neutral firms, they suffer more from the shock than neutral firms. We note this effect occurs despite the fact that overconfident firms tend to rely more on internal funds and less on external funds. The arbitrage between debt and equity is predominant relative to the arbitrage between internal and external funds.

Regarding the previous shock, we see how external decisions affect firms' capital structure and investment decisions. In this case, the monetary tightening delivers its effect by reducing investment; however, it affects firms asymmetrically. Neutral firms that rely more on the equity market suffer less from the shock.

## 6.3. More liquidity for firms

Finally, we consider the case of a 10% increase in firms' desired liquidity ratio; see (18) and (19). More precisely, we assume that at $t = 600$ all firms increase the lower and upper bounds of their target expected deposits to expenditures ratio from 0.1 to 0.11 and 0.5 to 0.55, respectively. The results are presented in Fig 9 and Table 5.

We see that higher liquidity holdings have a strong positive impact on the economy, as it significantly decreases firms' default rate (panel 1). Indeed, higher deposits allow firms to be more resilient to unexpected changes in demand, interest rate or real wages that would otherwise affect their profits and could push firms to default due to illiquidity. This lower default rate eases credit conditions, as banks suffer fewer losses due to firms failing to repay their loans. However, we do not observe a decrease in the interest rate (panel 6). Rather, we see that firms use this new financial space to increase their indebtedness (see panel 4). The interest rate remains roughly constant, as banks respond to a higher leverage by raising the interest rate (see 79). These new loans are used mostly to finance new investment, as we see a significant decrease in the utilization rate (panel 3) and only a moderate decrease in equity (panel 5).

The change in the liquidity ratio is slightly more efficient for neutral firms, as their default rate decreases more than that of overconfident firms, as shown in Table 5. This is because even with a very high liquidity ratio, overconfident firms are still quite likely to face illiquidity due to the bias in their profit expectations. By overestimating their future incoming cash flows,

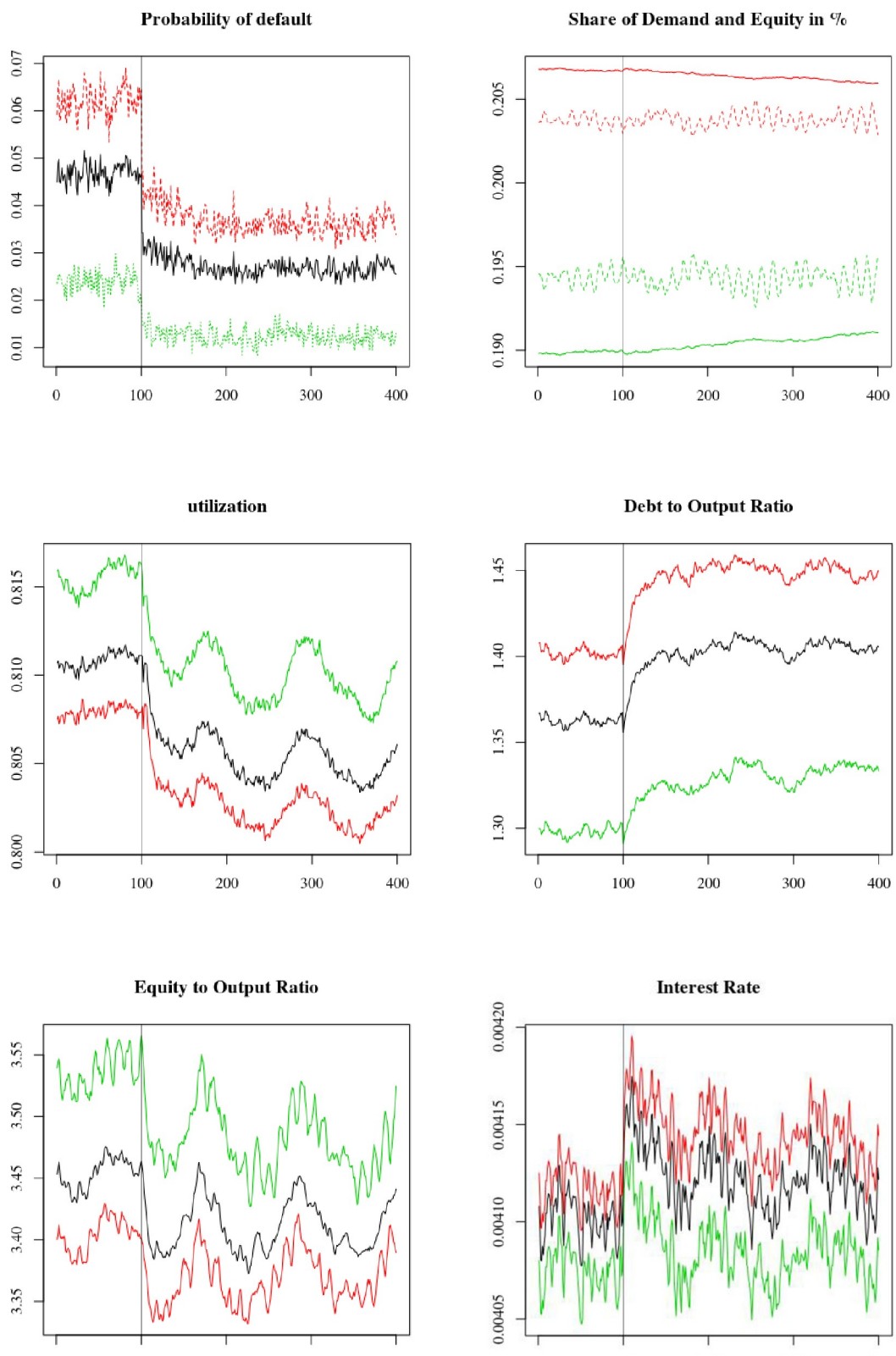

**Fig 9. Firms' liquidity ratio: Increase in the parameters $\lambda_{d,min}$ and $\lambda_{d,max}$ at $t = 600$.** Average firms (black), overconfident firms (red) and neutral firms (green).

Table 5. Shock on parameters $\lambda_{d,min}$ and $\lambda_{d,max}$.

|  | Default | Demand | Mkt. Cap. | Utilization | Debt | Equity | $i_l$ |
|---|---|---|---|---|---|---|---|
| Average | -42.9 | 0 | 0 | -0.74 | 3.17 | -1.35 | 0.46 |
|  | (0.18) | (3.30) | (1.96) | (0.27) | (0.46) | (0.52) | (1.24) |
| Overconfident | -41.2 | -0.39 | 0 | -0.75 | 3.43 | -1.05 | 0.61 |
|  | (0.20) | (1.34) | (95.5) | (0.46) | (0.51) | (1.32) | (0.98) |
| Neutral | -48.3 | 0.65 | 0.01 | -0.76 | 2.76 | -1.82 | 0.23 |
|  | (0.26) | (1.33) | (94.3) | (0.34) | (0.65) | (0.93) | (3.04) |

Average variation of probability (Default), share of demand (Demand), share of total market capitalization (Mkt. Cap.), utilization, debt to output ratio (Debt), equity to output ratio (Equity) and interest rate ($i_l$) between steady state before and after the shock, in %. Coefficients of variation are in parentheses.

these firms tend to use their internal funds excessively to finance investment, causing a default when the cash flows that were supposed to replenish their treasury never arrive. Nonetheless, a higher liquidity ratio remains a highly efficient measure to stabilize the economy, as it significantly decreases the probability of default for all firms, even the overconfident ones. The higher liquidity ratio being more effective for neutral firms, we see that they manage to increase their equity and market share at the expense of overconfident firms.

Here, we observe results with a Minskian flavour where an increase in cushions of safety by firms tends to stabilize the economy and lead to higher investments. Within this overall stabilization process, we see that more prudent firms do better than more overconfident ones.

## 7. Conclusion

This paper contributes to the knowledge base by linking behavioural corporate finance and agent-based macroeconomics. The main goal was to understand how overconfident behaviours related to the capital structure of a specific firm could affect that firm's dynamics, the financial market, macroeconomic dynamics and other firms' dynamics via the macroeconomic environment. The approach is novel in terms of at least two aspects. We explicitly account for different behavioural rules and embed these in a macroeconomic framework, thus allowing us to track the impact at the micro and macro levels. On the other hand, we micro-calibrate our model using the results of a behavioural finance survey in addition to replicating macroeconomic stylized facts.

Our results also highlight the importance to look at micro and macro dynamics, as the increased level of debt demonstrated by overconfident firms does not lead to an overall increase in debt level. Firms that are more indebted are also more fragile and force banks to change their interest rate to compensate for losses, leading to an overall decrease in debt level. It would have been interesting to understand how these mechanisms impact the banking network and see if one would observe avalanches of bankruptcies, along the line of Delli Gatti et al., (2010) and Riccetti, Russo & Gallegati (2013). This is left for further work.

The current framework ignores dynamics associated with the determinants of productivity and rather assumes exogenous labour productivity growth. This simplifying assumption constrains the stationary state of the economy, as the long-term growth rate of output has to follow the productivity trend. Combining insights from behavioural corporate finance, such as excess overconfidence, with the literature on endogenous growth and innovation is an interesting research avenue that we leave to future studies. We nonetheless see that excess overconfidence, either an increase in the proportion of overconfident firms or a higher degree of overconfidence, has a strong destabilizing impact on the economy. Greater reliance on internal funds and bank loans increases firms' probability of default, which in turn negatively affects the

banking sector and all the economy, as highlighted by the higher volatility of output growth and return on equity or the lower use of banking credit by firms.

Finally, we modelled three policy shocks, one in the form of monetary policy tightening, one in the form of a flight to liquidity and one in the form of more liquid firms. We see that the monetary policy delivers the expected outcome via an increase in interest rates and a decrease in investment. Reflecting a loss in confidence by financial investors, the flight to liquidity has important effects in the form of short-run financial volatility, an increased default rate and lower investment in the long run. Our model indicates that more overconfident firms displaying higher investment propensities are also more fragile. The third policy analysis shows that this can be offset by having larger cushions of safety in the form of holding more liquid assets. This allows compensating for the overconfidence, leading to an increase in investment for the entire economy. In the current situation of financial turmoil, it therefore seems interesting to implement counter-cyclical macro-prudential policies where firms are encouraged to hold more liquid assets to better absorb financial fluctuations.

## Supporting information

**S1 Table. Parameter values.**
(DOCX)

**S1 File.**
(SAV)

## Author Contributions

**Conceptualization:** Marcin Rzeszutek, Antoine Godin, Adam Szyszka.

**Data curation:** Antoine Godin, Stanislas Augier.

**Formal analysis:** Marcin Rzeszutek, Antoine Godin, Stanislas Augier.

**Funding acquisition:** Marcin Rzeszutek, Adam Szyszka.

**Investigation:** Adam Szyszka.

**Methodology:** Marcin Rzeszutek, Antoine Godin.

**Project administration:** Marcin Rzeszutek, Stanislas Augier.

**Resources:** Adam Szyszka, Stanislas Augier.

**Supervision:** Marcin Rzeszutek.

**Writing – original draft:** Marcin Rzeszutek, Antoine Godin.

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
