## [Decision Letter · Decision Letter 0]

1 Jul 2021

PONE-D-21-17371

Managerial overconfidence in capital structure decisions and its link to aggregate demand: an agent-based model perspective

PLOS ONE

Dear Dr. Rzeszutek,

Thank you for submitting your manuscript to PLOS ONE. After careful consideration, we feel that it has merit but does not fully meet PLOS ONE’s publication criteria as it currently stands. 

The manuscript is very near of being suitable of publication. One of the reviewers have some minor concerns that need to be addressed in a revised version.  

We look forward to receiving your revised manuscript.

Kind regards,

J E. Trinidad Segovia

Academic Editor

PLOS ONE

Journal Requirements:

1. Please ensure that your manuscript meets PLOS ONE's style requirements, including those for file naming. The PLOS ONE style templates can be found athttps://journals.plos.org/plosone/s/file?id=wjVg/PLOSOne_formatting_sample_main_body.pdf and https://journals.plos.org/plosone/s/file?id=ba62/PLOSOne_formatting_sample_title_authors_affiliations.pdf

'The funders had no role in study design, data collection and analysis, decision to

publish, or preparation of the manuscript.'

Additional Editor Comments (if provided):

Reviewers' comments:

Reviewer's Responses to Questions

**Comments to the Author**

1. Is the manuscript technically sound, and do the data support the conclusions?

Reviewer #1: Yes

Reviewer #2: Yes

2. Has the statistical analysis been performed appropriately and rigorously? 

Reviewer #1: Yes

Reviewer #2: Yes

3. Have the authors made all data underlying the findings in their manuscript fully available?

Reviewer #1: Yes

Reviewer #2: Yes

4. Is the manuscript presented in an intelligible fashion and written in standard English?

Reviewer #1: Yes

Reviewer #2: Yes

5. Review Comments to the Author

Reviewer #1: The article addresses the impact of managerial overconfidence at the micro and macro levels of the economy as a whole. The study combines agent-based macroeconomics and behavioural finance. It is interesting and covers a wide range of aspects. Below, I include some comments that can help the Authors to improve their article.

1. On page 2, there is the objective ‘This study aims to connect two strands of the psychology and economics literature, i.e., behavioural finance and agent-based macroeconomics, to assess the impact of managerial overconfidence at the micro and macro levels of the economy as a whole.’

But later, the aim is stated this way:

‘The aim here is to investigate the two-way association between managerial practices relating to the target capital structure of a company and selected macroeconomic indicators at the aggregate level.’

In my opinion, it would be more beneficial to choose precisely one sentence which is called either objective or aim and always refer to this one sentence (which of course, can be explained by some further sentences in the paragraph).

2. There are some inconsistencies when it comes to the mathematical approach. Some variables or indices are weakly explained or not explained at all, and the reader must guess them from the text. For example, is ‘i’ is for the number of firms? It is not explained. And ki in the third equation on page 10 is explained much later on page 12. The next unexplained variable is λ , if it was at least added in the text next to the description, it would be more comprehensible. In equation (1), the variable on the left side is ydei. But, on the right side, there is ydi,-1. If the second subscript is added, then the main variable on the left side of the equation should be ydei,t. Moreover, if it is about time, and we refer to a given time, so I would recommend to use ‘t-1’ instead of ‘-1’. In my opinion, it would be beneficial to consult all the formulas with a mathematician.

3. The way of referring to the equations is a bit odd, and it should be corrected. In this statement: ‘C-Firms’ expected demand (, 1) follows a double…’, it seems like the Authors were referring to a vector function of the expected demand. Generally, in mathematics, when referring to a given equation, only the number of the equation is shown in parentheses. So, it should be ‘C-Firms’ expected demand (1) follows a double…’, and later ‘C-firms have a target inventories to expected demand ratio (2), following…’ and the same for each reference to each equation.

4. There is some unnecessary numbering at the beginning of chapter 3.

5. On page 4, the last sentence ‘C-firms expected sales are bounded by their production decision plus available inventories (3)’. It should be ‘4’ instead of ‘3’, or to be consistent with the other references ‘(e, 4)’.

6. In the article, there are many equations, but not all of them are directly referred to in the text. In my opinion, it would be more consistent, and it would help the reader, if the reference to each equation was added (see for instance the equations 33, 34, 45)

7. Subchapter 1.1. There is only one subchapter at this level. So, I would recommend choosing one of these two options:

a) Move it to chapter 1 (without a separate 1.1 subchapter)

b) Move the first part of this subchapter considering the current study to chapter 2, and move the information about the remainder of the paper to chapter no. 1.

I leave the decision to the Authors.

8. After subchapter 3.1. the next is 8.1 – please correct the numbering.

9. Please think about what this study is about. Is it optimism that affects beliefs concerning the probability of a payoff, or is it overconfidence that affects beliefs when individuals assess their own performance? Referring to formulas on page 12, I would say that it is overconfidence.

Overconfidence seemed to be a focal term in this study, and it is exposed in the title but not in the text. In subchapter 8.1, the Authors refer to ‘optimistic’ firms. I would recommend referring to ‘overconfidence’ instead to strengthen the importance of this term in the study (from page 8 to page 34, ‘overconfidence’ is not mentioned). Similarly, it should be emphasised in subchapter 10.2 (pp. 26-27) and in the Conclusions section (where it is mentioned only once).

10. On page 10, the Authors refer to Steindl (1952) and Lavoie (1992). What about the latter formulas? Are they the Authors own work? Aren’t they based on any other studies? There are no other references till page 19. If they are the Authors own work, it should be emphasised in the text.

11. On page 20, in descriptions of equations (65-67), there is a phonetic name ‘rho’, but in the equation, there is a letter ρ, e.g., ρ_i^e=ρ ®_i. It would be beneficial to use just the letter ρ in order not to confuse the readers. Moreover, it would be consistent with the other descriptions.

12. p. 22, unnecessary square bracket: ‘…have enough own funds (82]) to satisfy…’

13. p. 25: ‘The values for all parameters are in Table 3, in Appendix.’ Shouldn’t it be Table 6? However, this table is quite long. So, it could be moved to an Appendix.

14. In the title of figure 1, it would be beneficial to add the information that it refers to the structure of GDP.

15. Please check the language before resubmitting, e.g., p. 25 ‘Figure 2 display the cyclical components of GDP….’

16. The formatting needs to be improved following the standards for PLOS ONE articles: https://journals.plos.org/plosone/s/submission-guidelines

Reviewer #2: The authors present an agent-based model of macroeconomics seeking to investigate the impact of managerial overconfidence on the micro and macro levels of the economy.

The proposed agent-based model links micro and macro parts of the analysis, helping to combine the behavioral aspect at the micro-level with the aggregated results at the macro level. The proposed model is calibrated for the specific macro indicators of Poland and uses recent findings of previous studies. The main idea is beautiful as it modifies the neoclassical approach by introducing the agent heterogeneity relating it to the target capital structure. Though there is a feeling that authors investigate properties of artificially constructed dynamical system witch has fragile relation to the real economy under consideration, the qualitative interpretation of the model behavior and numerical results support the made conclusions.

Readers of this manuscript have to look at the many cited resources to follow the authors' thoughts. Nevertheless, the model description looks very detailed and accurate. We would propose this manuscript for publication in Plos One.

PS: There are left typos in the manuscript, e.g., the same sentence is repeated twice in pages 27-28.

6. PLOS authors have the option to publish the peer review history of their article (what does this mean?). If published, this will include your full peer review and any attached files.

Reviewer #1: No

Reviewer #2: No

---

## [Author Response · Author response to Decision Letter 0]

5 Jul 2021

Dear Editor, Dear Reviewers, 

thank you very much for your suggestions and remarks concerning our article titled “Managerial overconfidence in capital structure decisions and its link to aggregate demand: an agent-based model perspective”, which we would like to publish in PLOS One. Below we cite the every remark and comment and provide our answers to them in parentheses. All the changes in the text are marked with the red font. 

Editor statements

[Yes, our manuscript meets PLOS ONE's style requirements mentioned above.]

'The funders had no role in study design, data collection and analysis, decision to

publish, or preparation of the manuscript.At this time, please address the following queries:

a. Please clarify the sources of funding (financial or material support) for your study. List the grants or organizations that supported your study, including funding received from your institution.

d. If you did not receive any funding for this study, please state: “The authors received no specific funding for this work.”Please include your amended statements within your cover letter; we will change the online submission form on your behalf.

[Thank you for this clarification. We included all the details regarding the funding as is mentioned above.]

[Also the reference list was again reviewed to check and verify its correctness.]

Reviewers' comments:

Reviewer's Responses to Questions

Comments to the Author

1. Is the manuscript technically sound, and do the data support the conclusions?

Reviewer #1: Yes

Reviewer #2: Yes

2. Has the statistical analysis been performed appropriately and rigorously?

Reviewer #1: Yes

Reviewer #2: Yes

3. Have the authors made all data underlying the findings in their manuscript fully available?

Reviewer #1: Yes

Reviewer #2: Yes

4. Is the manuscript presented in an intelligible fashion and written in standard English?

Reviewer #1: Yes

Reviewer #2: Yes

[Overall, thank you both reviewers for positive, quantitative feedback on our manuscript.]

Reviewer #1: The article addresses the impact of managerial overconfidence at the micro and macro levels of the economy as a whole. The study combines agent-based macroeconomics and behavioural finance. It is interesting and covers a wide range of aspects. Below, I include some comments that can help the Authors to improve their article.

1. On page 2, there is the objective ‘This study aims to connect two strands of the psychology and economics literature, i.e., behavioural finance and agent-based macroeconomics, to assess the impact of managerial overconfidence at the micro and macro levels of the economy as a whole.’ But later, the aim is stated this way:

‘The aim here is to investigate the two-way association between managerial practices relating to the target capital structure of a company and selected macroeconomic indicators at the aggregate level.’

In my opinion, it would be more beneficial to choose precisely one sentence which is called either objective or aim and always refer to this one sentence (which of course, can be explained by some further sentences in the paragraph).

[Thank you very much for this remark. It is true that this one sentence is more communicative, and thus, we changed it to one sentence in accordance to Reviewer’s 1 suggestion.]

2. There are some inconsistencies when it comes to the mathematical approach. Some variables or indices are weakly explained or not explained at all, and the reader must guess them from the text. For example, is ‘i’ is for the number of firms? It is not explained. And ki in the third equation on page 10 is explained much later on page 12. The next unexplained variable is λ , if it was at least added in the text next to the description, it would be more comprehensible. In equation (1), the variable on the left side is ydei. But, on the right side, there is ydi,-1. If the second subscript is added, then the main variable on the left side of the equation should be ydei,t. Moreover, if it is about time, and we refer to a given time, so I would recommend to use ‘t-1’ instead of ‘-1’. In my opinion, it would be beneficial to consult all the formulas with a mathematician.

[Thank you very much for this remark. It is true that we missed some clarifications in describing the equations in a consistent way. We made sure to explain every variable appearing in the equations while identifying parameters. In this way, we hope that the reading of the text is now more fluid and clearer.]

3. The way of referring to the equations is a bit odd, and it should be corrected. In this statement: ‘C-Firms’ expected demand (, 1) follows a double…’, it seems like the Authors were referring to a vector function of the expected demand. Generally, in mathematics, when referring to a given equation, only the number of the equation is shown in parentheses. So, it should be ‘C-Firms’ expected demand (1) follows a double…’, and later ‘C-firms have a target inventories to expected demand ratio (2), following…’ and the same for each reference to each equation.

[Thank you again for paying our attention to our inconsistency. In the revised version of manuscript, we now propose both the equation number and the name of the variable.]

4. There is some unnecessary numbering at the beginning of chapter 3.

[Thank you for this remark. We corrected it.]

5. On page 4, the last sentence ‘C-firms expected sales are bounded by their production decision plus available inventories (3)’. It should be ‘4’ instead of ‘3’, or to be consistent with the other references ‘(e, 4)’.

[Again – thank you very much for catching all our formal mistakes. We apology for them. Obviously, we corrected it.]

6. In the article, there are many equations, but not all of them are directly referred to in the text. In my opinion, it would be more consistent, and it would help the reader, if the reference to each equation was added (see for instance the equations 33, 34, 45).

[Thank for this remark. Yes, we also think that it is a good idea and we did update the text accordingly]

7. Subchapter 1.1. There is only one subchapter at this level. So, I would recommend choosing one of these two options:

a) Move it to chapter 1 (without a separate 1.1 subchapter)

b) Move the first part of this subchapter considering the current study to chapter 2, and move the information about the remainder of the paper to chapter no. 1.

I leave the decision to the Authors.

[The best idea for us was a. We moved it to chapter 1 without a separate 1.1 subchapter.]

8. After subchapter 3.1. the next is 8.1 – please correct the numbering.

[We corrected it, Thank you.]

9. Please think about what this study is about. Is it optimism that affects beliefs concerning the probability of a payoff, or is it overconfidence that affects beliefs when individuals assess their own performance? Referring to formulas on page 12, I would say that it is overconfidence.

Overconfidence seemed to be a focal term in this study, and it is exposed in the title but not in the text. In subchapter 8.1, the Authors refer to ‘optimistic’ firms. I would recommend referring to ‘overconfidence’ instead to strengthen the importance of this term in the study (from page 8 to page 34, ‘overconfidence’ is not mentioned). Similarly, it should be emphasised in subchapter 10.2 (pp. 26-27) and in the Conclusions section (where it is mentioned only once).

[Thank you very much for this fundamental remark. Yes, the core psychological bias we assessed is overconfidence. We used “optimism” word to enhance the flow of the text, as synonym. But we now stick to “overconfidence” throughout 

10. On page 10, the Authors refer to Steindl (1952) and Lavoie (1992). What about the latter formulas? Are they the Authors own work? Aren’t they based on any other studies? There are no other references till page 19. If they are the Authors own work, it should be emphasised in the text.

[Thank you for this remark, we were indeed a bit light on the references. We made sure to highlight where we were inspired by specific articles or where we used exact formulations from the literature.]

11. On page 20, in descriptions of equations (65-67), there is a phonetic name ‘rho’, but in the equation, there is a letter ρ, e.g., ρ_i^e=ρ ®_i. It would be beneficial to use just the letter ρ in order not to confuse the readers. Moreover, it would be consistent with the other descriptions.

12. p. 22, unnecessary square bracket: ‘…have enough own funds (82]) to satisfy…’

13. p. 25: ‘The values for all parameters are in Table 3, in Appendix.’ Shouldn’t it be Table 6? However, this table is quite long. So, it could be moved to an Appendix.

[Thank you for these remarks, we have indeed corrected all these typos]

14. In the title of figure 1, it would be beneficial to add the information that it refers to the structure of GDP.

[Very good suggestion. We did accordingly.]

15. Please check the language before resubmitting, e.g., p. 25 ‘Figure 2 display the cyclical components of GDP….’

[We sent our submission tot the professional English editing service. However, we now again checked the text to look for some inconsistencies, like above.]

16. The formatting needs to be improved following the standards for PLOS ONE articles: https://journals.plos.org/plosone/s/submission-guidelines

[Thank you for this remark. We adhered to PLOS one formatting.]

Reviewer #2: The authors present an agent-based model of macroeconomics seeking to investigate the impact of managerial overconfidence on the micro and macro levels of the economy. The proposed agent-based model links micro and macro parts of the analysis, helping to combine the behavioral aspect at the micro-level with the aggregated results at the macro level. The proposed model is calibrated for the specific macro indicators of Poland and uses recent findings of previous studies. The main idea is beautiful as it modifies the neoclassical approach by introducing the agent heterogeneity relating it to the target capital structure. Though there is a feeling that authors investigate properties of artificially constructed dynamical system witch has fragile relation to the real economy under consideration, the qualitative interpretation of the model behavior and numerical results support the made conclusions.

Readers of this manuscript have to look at the many cited resources to follow the authors' thoughts. Nevertheless, the model description looks very detailed and accurate. We would propose this manuscript for publication in Plos One.

PS: There are left typos in the manuscript, e.g., the same sentence is repeated twice in pages 27-28.

[Thank you very much for this very kind words on our manuscript. We highly appreciate them. Finally, we corrected the aforementioned typo.]

To sum up, we would like to thank again the Editor and Reviewers for suggestions and remarks concerning our manuscript. We found these comments very useful as they provided us with help and guidance on how to improve quality of the manuscript. We really appreciated the chance you gave us to revise and resubmit our manuscript to PLOS One.

---

## [Decision Letter · Decision Letter 1]

13 Jul 2021

PONE-D-21-17371R1

Managerial overconfidence in capital structure decisions and its link to aggregate demand: an agent-based model perspective

PLOS ONE

Dear Dr. Rzeszutek,

Thank you for submitting your manuscript to PLOS ONE. After careful consideration, I feel that it has merit but does not fully meet PLOS ONE’s publication criteria as it currently stands. 

Most of the major comments have been addressed but unfortunately there are some minor questions that need to be solved before publication. I suggest to the authors to attend this questions a re-submit the revised version as soon as possible.

We look forward to receiving your revised manuscript.

Kind regards,

J E. Trinidad Segovia

Academic Editor

PLOS ONE

Journal Requirements:

Reviewers' comments:

Reviewer's Responses to Questions

**Comments to the Author**

1. If the authors have adequately addressed your comments raised in a previous round of review and you feel that this manuscript is now acceptable for publication, you may indicate that here to bypass the “Comments to the Author” section, enter your conflict of interest statement in the “Confidential to Editor” section, and submit your "Accept" recommendation.

Reviewer #1: (No Response)

2. Is the manuscript technically sound, and do the data support the conclusions?

Reviewer #1: Yes

3. Has the statistical analysis been performed appropriately and rigorously? 

Reviewer #1: Yes

4. Have the authors made all data underlying the findings in their manuscript fully available?

Reviewer #1: Yes

5. Is the manuscript presented in an intelligible fashion and written in standard English?

Reviewer #1: Yes

6. Review Comments to the Author

Reviewer #1: Thank you for your efforts to correct the article. I noticed that the issues I raised were generally well addressed. Here are just the last comments and remarks:

1. On p. 8, the Authors wrote ‘firm’s overconfidence’. Wouldn’t it be more precise to write ‘managerial overconfidence’? I leave this decision to the Authors.

2. Subchapter 3.1. starts with 5 points. In my opinion, this numbering confuses the reader, especially if there are also numbers of equations that are different from numbering. I think it could be deleted without losing the clarity of the text. I leave this decision to the Authors.

3. I can see that in my previous review, some variables are missing. So, I will write once again what I meant to say. The way of referring to the equations is a bit odd, and in my opinion, should be corrected. In this statement: ‘C-Firms’ expected demand (ydei, 1) follows a double….’ It seems like the Authors were referring to a vector function of the expected demand. Generally, in mathematics, when referring to a given equation, only the number of the equation is shown in parentheses (the variable is not). So, I recommend referring to formulas this way: ‘C-Firms’ expected demand ydei (1) follows a double…’. In my opinion this change would be beneficial. However, I leave this decision to the Authors.

4. On p. 12, the expression in brackets was added. It explains the formulas sufficiently. However, there are some hyphens that may confuse the reader. ‘(composed of wages, interests payments on debt -, ⋅ ,−1- and interests receipts on deposits -, ⋅ ,−1-)’. The Authors could discuss it, and consider changing the text in parentheses.

5. Appendix starts with:

‘Appendix

[INSERT TABLE 3 ABOUT HERE]

Table 1: Balance sheet...’

Please correct it.

7. PLOS authors have the option to publish the peer review history of their article (what does this mean?). If published, this will include your full peer review and any attached files.

Reviewer #1: No

---

## [Author Response · Author response to Decision Letter 1]

13 Jul 2021

Dear Editor, Dear Reviewers, 

thank you very much for another suggestions and remarks concerning our article titled “Managerial overconfidence in capital structure decisions and its link to aggregate demand: an agent-based model perspective”, which we would like to publish in PLOS One. Below we cite every remark and comment and provide our answers to them in parentheses. All the changes in the text are marked with the red font. 

Reviewer Thank you for your efforts to correct the article. I noticed that the issues I raised were generally well addressed. Here are just the last comments and remarks:

[Thank you very much for kind words on our revision.]

1. On p. 8, the Authors wrote ‘firm’s overconfidence’. Wouldn’t it be more precise to write ‘managerial overconfidence’? I leave this decision to the Authors.

[Thank you very much for this remark - we wrote ‘managerial overconfidence’.]

2. Subchapter 3.1. starts with 5 points. In my opinion, this numbering confuses the reader, especially if there are also numbers of equations that are different from numbering. I think it could be deleted without losing the clarity of the text. I leave this decision to the Authors.

[Thank you very much for another interesting suggestion -we did as the reviewer suggested.]

3. I can see that in my previous review, some variables are missing. So, I will write once again what I meant to say. The way of referring to the equations is a bit odd, and in my opinion, should be corrected. In this statement: ‘C-Firms’ expected demand (ydei, 1) follows a double….’ It seems like the Authors were referring to a vector function of the expected demand. Generally, in mathematics, when referring to a given equation, only the number of the equation is shown in parentheses (the variable is not). So, I recommend referring to formulas this way: ‘C-Firms’ expected demand ydei (1) follows a double…’. In my opinion this change would be beneficial. However, I leave this decision to the Authors.

[Thank you very much for this remark. However, according to the mathematician we work with, the way of referring to equation was done in purpose. This reflects the ABM simulation style and it was generally following the style adopted in this field. But thank you for your suggestion again.]

4. On p. 12, the expression in brackets was added. It explains the formulas sufficiently. However, there are some hyphens that may confuse the reader. ‘(composed of wages, interests payments on debt -𝑖𝑙,𝑖 ⋅ 𝑙𝑖,−1- and interests receipts on deposits -, 𝑖𝑑 ⋅ 𝑑𝑖,−1-)’. The Authors could discuss it, and consider changing the text in parentheses.

[Again, we would like to underscore that it was done in purpose - according to the mathematician we work with, the way of referring to equation was done in purpose. This reflects the ABM simulation style and it was generally following the style adopted in this field. But thank you for your suggestion again.]

5. Appendix starts with:

‘Appendix

[INSERT TABLE 3 ABOUT HERE]

Table 1: Balance sheet...’

Please correct it.

[Thank you -we corrected it.]

To sum up, we would like to thank again the Editor and Reviewers for suggestions and remarks concerning our manuscript. We found these comments very useful as they provided us with help and guidance on how to improve quality of the manuscript. We really appreciated the chance you gave us to revise and resubmit our manuscript to PLOS One.

---

## [Decision Letter · Decision Letter 2]

19 Jul 2021

Managerial overconfidence in capital structure decisions and its link to aggregate demand: an agent-based model perspective

PONE-D-21-17371R2

Dear Dr. Rzeszutek,

We’re pleased to inform you that your manuscript has been judged scientifically suitable for publication and will be formally accepted for publication once it meets all outstanding technical requirements.

Kind regards,

J E. Trinidad Segovia

Section Editor

PLOS ONE

Additional Editor Comments (optional):

Reviewers' comments:

Reviewer's Responses to Questions

**Comments to the Author**

1. If the authors have adequately addressed your comments raised in a previous round of review and you feel that this manuscript is now acceptable for publication, you may indicate that here to bypass the “Comments to the Author” section, enter your conflict of interest statement in the “Confidential to Editor” section, and submit your "Accept" recommendation.

Reviewer #1: All comments have been addressed

2. Is the manuscript technically sound, and do the data support the conclusions?

Reviewer #1: Yes

3. Has the statistical analysis been performed appropriately and rigorously? 

Reviewer #1: Yes

4. Have the authors made all data underlying the findings in their manuscript fully available?

Reviewer #1: Yes

5. Is the manuscript presented in an intelligible fashion and written in standard English?

Reviewer #1: Yes

6. Review Comments to the Author

Reviewer #1: All my comments were adequately addressed. In my opinion, the article can be accepted for publication.

7. PLOS authors have the option to publish the peer review history of their article (what does this mean?). If published, this will include your full peer review and any attached files.

Reviewer #1: No

---

## [Editor Report · Acceptance letter]

28 Jul 2021

PONE-D-21-17371R2 

Managerial overconfidence in capital structure decisions and its link to aggregate demand: an agent-based model perspective 

Dear Dr. Rzeszutek:

I'm pleased to inform you that your manuscript has been deemed suitable for publication in PLOS ONE. Congratulations! Your manuscript is now with our production department. 

Kind regards, 

on behalf of

Dr. J E. Trinidad Segovia 

Section Editor

PLOS ONE